# Calpain fosters the hyperexcitability of motoneurons after spinal cord injury and leads to spasticity

**Vanessa Plantier[†], Irene Sanchez-Brualla[†], Nejada Dingu, Cécile Brocard, Sylvie Liabeuf, Florian Gackière, Frédéric Brocard***

Institut de Neurosciences de la Timone (UMR7289), Aix-Marseille Université and CNRS, Marseille, France

**Abstract** Up-regulation of the persistent sodium current ($I_{NaP}$) and down-regulation of the potassium/chloride extruder KCC2 lead to spasticity after spinal cord injury (SCI). We here identified calpain as the driver of the up- and down-regulation of $I_{NaP}$ and KCC2, respectively, in neonatal rat lumbar motoneurons. Few days after SCI, neonatal rats developed behavioral signs of spasticity with the emergence of both hyperreflexia and abnormal involuntary muscle contractions on hindlimbs. At the same time, in vitro isolated lumbar spinal cords became hyperreflexive and displayed numerous spontaneous motor outputs. Calpain-I expression paralleled with a proteolysis of voltage-gated sodium (Nav) channels and KCC2. Acute inhibition of calpains reduced this proteolysis, restored the motoneuronal expression of Nav and KCC2, normalized $I_{NaP}$ and KCC2 function, and curtailed spasticity. In sum, by up- and down-regulating $I_{NaP}$ and KCC2, the calpain-mediated proteolysis of Nav and KCC2 drives the hyperexcitability of motoneurons which leads to spasticity after SCI.

**\*For correspondence:**
frederic.brocard@univ-amu.fr

[†]These authors contributed equally to this work

**Competing interests:** The authors declare that no competing interests exist.

## Introduction

The main clinical symptoms of spasticity, hyperreflexia and spasms, develop after spinal cord injury (SCI) due in part to the hyperexcitability of motoneurons (*Hultborn, 2003*; *Gorassini, 2004*; *Nielsen et al., 2007*; *D'Amico et al., 2014*; *Thomas et al., 2014*). Understanding the cellular pathophysiological processes underlying this hyperexcitability might offer new therapeutic perspectives for spasticity.

SCI enhances the intrinsic excitability of motoneurons by upregulating their persistent sodium ($I_{NaP}$) and calcium ($I_{CaP}$) currents, leading to muscle spasms and hyperreflexia in both humans and adult rats (*Li and Bennett, 2003*; *Li et al., 2004*; *Harvey et al., 2005*; *Harvey et al., 2006b*; *Harvey et al., 2006a*; *ElBasiouny et al., 2010*; *Theiss et al., 2011*; *Brocard et al., 2016*). In addition, a concomitant synaptic disinhibition of motoneurons due to a decrease of the main chloride extruder KCC2 also takes place after SCI in both humans and adult rodents (*Boulenguez et al., 2010*; *Mòdol et al., 2014*; *Klomjai et al., 2019*). In vitro experiments from neonatal rats show that this form of disinhibition stems primarily from an impaired Cl⁻ extrusion, typically identified by a depolarizing shift of the reversal potential of inhibitory postsynaptic potentials ($E_{IPSP}$) (*Boulenguez et al., 2010*), which may facilitate the recruitment of persistent inward currents after SCI (*Venugopal et al., 2011*). Thus, up-regulation of persistent inward currents concomitant with down-regulation of KCC2 may have a synergistic effect causing spasticity.

Remarkable advances have been made in the molecular mechanisms involved in alterations of persistent inward currents after SCI. SCI-induced constitutive 5-HT$_{2B/C}$ receptor activity leads to an increase in $I_{CaP}$ (*Murray et al., 2010*), while calpain-mediated proteolysis of Nav1.6 channels up-regulates $I_{NaP}$ (*Brocard et al., 2016*). Although the *mechanisms* involved in alterations of KCC2 after

SCI remain elusive, it is worth mentioning that calpain-mediated cleavage of KCC2 depolarizes the $E_{IPSP}$ in some pathophysiological conditions (*Puskarjov et al., 2012*; *Zhou et al., 2012*; *Wan et al., 2018*). Our study investigates whether SCI-induced activation of calpains is upstream of the *up- and down-regulation of $I_{NaP}$ and KCC2* in motoneurons after SCI. If so, we aim at demonstrating whether a cooperation between calpain-mediated alterations of $I_{NaP}$ and $E_{IPSP}$ is a necessary element driving spasticity. For this purpose, we characterized signs of spasticity from the neonatal rat SCI model to obtain easier in vitro correlates of the hyperexcitability of lumbar motoneurons controlling hindlimb muscles.

## Results

### Symptoms of spasticity in neonatal rats develop few days after SCI

Signs of spasticity (spontaneous muscle spasms, hyperreflexia) have been reported to emerge in adult rats weeks after a thoracic spinal transection (*Corleto et al., 2015*). Here, we identified signs of spasticity 4–5 days (d) after a thoracic transection performed within the first 12 hr (hrs) after birth in neonatal rats. Instead of a brief kicking of hindlimbs in response to the tail-pinch test (*Figure 1A* and *Figure 1—video 1*), rats with SCI displayed hindlimb hyperextension characterized by long-lasting EMG activity recorded from triceps surae extensor muscles (p<0.001; *Figure 1A–C* and *Figure 1—video 2*). In addition, the *threshold for response* to mechanical stimuli was significantly reduced (p<0.05; *Figure 1D*). Rats with SCI resting on a heated plate at 34°C (*Figure 1E*) showed a higher number of spontaneous muscle twitches from the tail and hindlimbs compared to sham-operated rats (p<0.001; *Figure 1F* and *Figure 1—videos 3–4*). In sum, behavioral indicators of spasticity emerge a few days after SCI in neonatal rats.

### Acute inhibition of calpain reduces spasticity symptoms in neonatal SCI rats

The activation of calpain contributes to spasticity in adult rats with chronic SCI (*Brocard et al., 2016*). We assessed whether a similar molecular mechanism exists in SCI neonatal rats. The acute intraperitoneal injection (i.p.) of MDL28170 at minimal or maximal effective doses for the reduction of calpain activity in the CNS (*Kawamura et al., 2005*; *Thompson et al., 2010*), dose-dependently reduced the number of spontaneous muscle twitches (p<0.001; *Figure 1G*). At the highest dose (120 mg/kg), twitches were less frequent up to 9 hr compared to vehicle-treated animals and temporarily dropped to levels similar to those found in intact animals over a period from 2 to 5 hr post-injection (*Figure 1G*, *dashed line*). Note that MDL28170 used at 120 mg/kg did not reduce spontaneous twitches in sham-operated animals suggesting that the effect of the drug was specific to rats with SCI and did not result from lethargy (p>0.05; *Figure 1—figure supplement 1*). In addition to spasms reduction, the acute inhibition of calpains with MDL28170 decreased the duration of EMG responses to tail-pinch in SCI neonatal rats (p<0.05, *Figure 1H*) without affecting the threshold to mechanical stimuli (p>0.05, *Figure 1H*). In sum, data revealed a causal relationship between the activation of calpain and the development of spasticity in the neonatal SCI model.

### The expression of Calpain-I increased after SCI

In the CNS, calpains exist in two major isoforms: μ-calpain (or calpain-I) and m-calpain (or calpain-II). To determine whether these calpains showed changes in their expression after SCI, we performed Western blots on lumbar spinal cord tissue isolated from 5-d old rats. Immunoblots revealed a SCI-induced increase in expression of calpain-I in both its forms (the 110 kDa inactive proenzyme and the 80 kDa active catalytic subunit; p<0.05; *Figure 2*). Expression of calpain-II did not change after SCI (p>0.05; *Figure 2*).

### Early appearance of hyperexcitability to caudal spinal cord after SCI

We investigated whether the in vivo emergence of spasticity correlated with an early appearance of hyperexcitability within the sublesional spinal cord. In isolated spinal cords from sham-operated rats, a supramaximal stimulation of the 5[th] lumbar (L5) dorsal root elicited a typical short-lasting response in the homologous ventral root (SLR, black traces; *Figure 3A–C*) with the presence of the monosynaptic reflex (arrows in insets; *Figure 3A–C*). The SLR (the early transient 40 ms response) did not

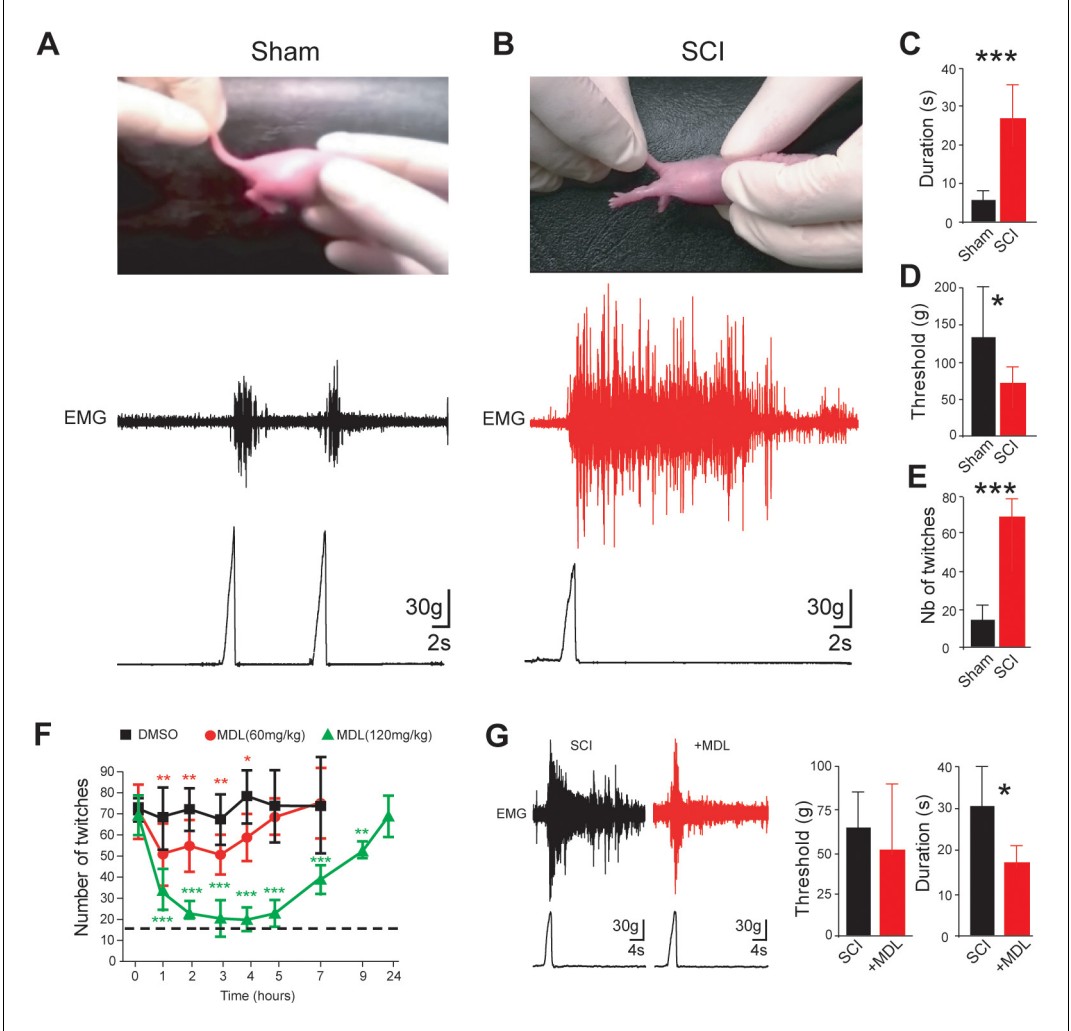

**Figure 1.** Calpain inhibition alleviates early behavioral signs of spasticity in neonatal rats with SCI. (**A,B**) Pictures of typical hindlimb motor response to tail pinching in sham-operated (**A**), n = 6 rats) and SCI rats (**B**), 5 d post-SCI, n = 8 rats). Lower and upper traces represent the pinch force and the evoked electromyographic (EMG) response of the triceps surae muscle, respectively. (**C,D**) Group means quantification of EMG responses. *p<0.05, ***p<0.001 comparing sham versus SCI groups; Mann-Whitney test. (**E**) Groups means quantification of twitches over a time period of 10 min obtained from sham-operated and SCI rats (5 d post-SCI) at rest on heating pad (~34.5℃). ***p<0.001 comparing sham versus SCI groups (n = 22 rats in both groups); Mann-Whitney test. (**F**) Time-course changes of twitches after acute i.p. administration (t = 0) of vehicle (black, n = 5 rats) or MDL28170 at 60 mg/kg (red, n = 5 rats) or MDL28170 at 120 mg/kg (green, n = 5 rats) in SCI neonatal rats (5 d post-SCI). *p<0.05, **p<0.01, ***p<0.001, repeated measures one-way ANOVA, Dunnett's post-test. (**G**) Representative EMG responses of the triceps surae muscle to tail pinching in SCI neonatal rats (5 d post-SCI) 60 min after an acute i.p. administration of vehicle (black, n = 9 rats) or MDL28170 (red, 120 mg/kg, n = 6 rats). Lower traces represent the pinch force. Group means quantification of EMG responses on the right-hand side. *p<0.05; comparing vehicle-treated versus MDL28170-treated rats; Mann-Whitney test. Underlying numerical values can be found in the *Figure 1—source data 1*.

The online version of this article includes the following video, source data, and figure supplement(s) for figure 1:

**Source data 1.** Values displayed in bar plots in *Figure 1C-G*.

**Figure supplement 1.** Calpain inhibition does not affect the number of twitches recorded in intact neonatal rats.

**Figure supplement 1—source data 1.** Values displayed in the time course shown in *Figure 1—figure supplement 1*.

**Figure 1—video 1.** Typical hindlimb motor movements to tail pinching in sham-operated neonatal rats.
https://elifesciences.org/articles/51404#fig1video1

**Figure 1—video 2.** Typical hindlimb motor movements to tail pinching in spinal cord transected neonatal rats.

*Figure 1 continued on next page*

*Figure 1 continued*

https://elifesciences.org/articles/51404#fig1video2

**Figure 1—video 3.** Typical spontaneous tail and hindlimb movements in sham-operated neonatal rats.
https://elifesciences.org/articles/51404#fig1video3
**Figure 1—video 4.** Typical spontaneous tail and hindlimb movements in spinal cord transected neonatal rats.
https://elifesciences.org/articles/51404#fig1video4

change regardless of age (p>0.05, in black, *Figure 3D–G*). By contrast, the SLR gradually increased after SCI (p<0.001, in red, *Figure 3D–G*) and a long-lasting reflex (LLR, red traces; *Figure 3A–C*), almost absent in sham-operated rats, was noticeable 24 hr post-surgery. Note that an interpulse interval of at least 4 min was required to avoid a use-dependent decline of the LLR (*Figure 3—figure supplement 1A–D*). The LLR gradually increased after SCI and differed significantly from sham-operated rats as early as 2–3 d post-SCI (p<0.001, *Figure 3H*), transforming the unimodal peristimulus time histograms (PSTHs) into a bimodal distribution (*Figure 3D–F*). One week after SCI, the LLR lasted 9.0 ± 0.6 s (n = 52 spinal cords, *Figure 3C*) and the stimulus threshold was markedly decreased (p<0.001, *Figure 3I*). At this age, in 16/52 (~31%) of our preparations, the LLR appeared as a fictive locomotor episode characterized by a rhythmic alternation and a negative cross-correlation between opposite ventral root activities (*Figure 3C*; *bottom traces*). This locomotor-like activity could not be evoked in sham-operated rats by a single pulse stimulation.

Stable spontaneous non-evoked motor bursts were also recorded from L5 ventral roots within the first hr after the spinal cord was placed in the recording chamber (*Figure 3J–L* and *Figure 3—figure supplement 1E–F*). The spontaneous motor bursts decreased in frequency during the first postnatal week in both SCI and sham-operated animals (black traces; *Figure 3J–M*), but were more frequent at all ages after SCI (red traces, p<0.001; *Figure 3J-M*). Spontaneous bursts also increased in both amplitude and duration with age after SCI, and were much more pronounced 4–5 d after SCI compared to sham-operated controls (p<0.001; *Figure 3N–O*). Sometimes spontaneous activities appeared as oscillatory bursts resembling locomotor-like episodes exclusively observed in SCI animals. Thus, an early increase of spinal hyperexcitability occurred after SCI and led to excessive spontaneous and sensory-evoked motor outputs consistent with behavioral signs of spasticity observed from SCI neonatal rats.

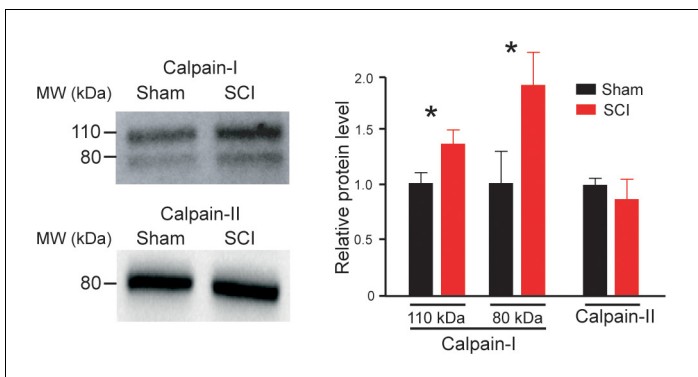

**Figure 2.** The expression of Calpain-I increased after SCI. *On the left*, calpain-I and calpain-II immunoblots of lumbar segments in sham-operated (*n* = 4 rats) and SCI neonatal rats (5 d post-SCI, *n* = 4 rats). One rat per lane. *On the right*, group means quantification of bands of both calpain-I and calpain-II in SCI rats normalized to sham-operated controls. *p<0.05 comparing sham versus SCI groups; Mann-Whitney test. Data are means ± SD. Underlying numerical values can be found in the *Figure 1—source data 1*.
The online version of this article includes the following source data for figure 2:

**Source data 1.** Values displayed in bar plots in *Figure 2*.

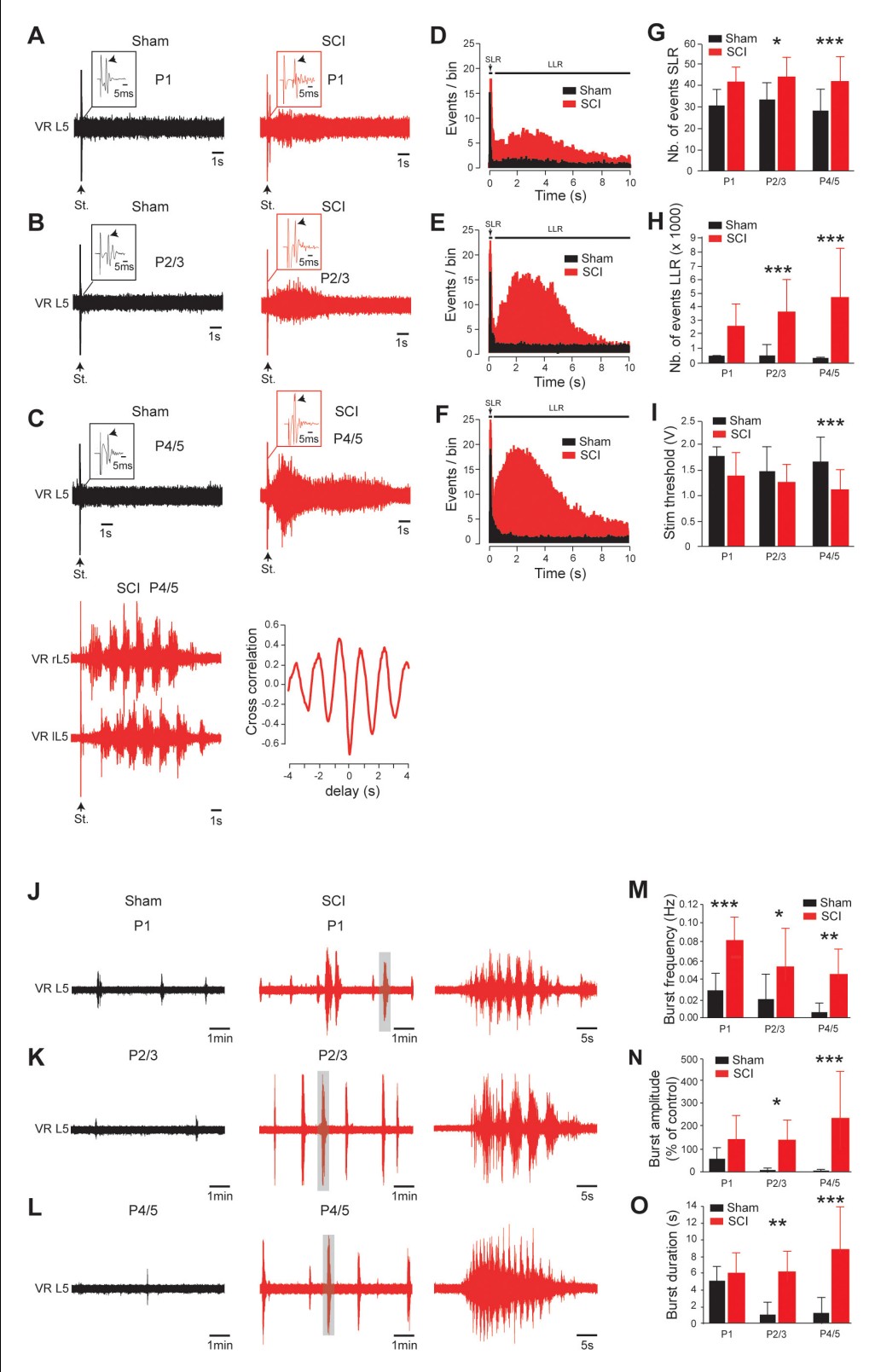

**Figure 3.** Early development of hyperexcitability on caudal spinal cord after SCI. (**A–C, J–L**) Representative L5 ventral root activities (VR L5) evoked by a supramaximal stimulation (St.) of the ipsilateral dorsal root (**A–C**) or occurring spontaneously (**J–L**) in spinal cords isolated 1 d (**A, J**), 2–3 d (**B,K**) or 4–5 d (**C,L**) after sham surgery (black) or SCI (red) [for **A–C**: 1 d, *n* = 24 sham and *n* = 6 SCI; 2–3 d, *n* = 29 sham and *n* = 10 SCI; 4–5 d, *n* = 52 in

*Figure 3 continued on next page*

*Figure 3 continued*

both groups; for **J–L**: 1 d, $n = 12$ sham and $n = 7$ SCI; 2–3 d, $n = 11$ sham and $n = 12$ SCI; 4–5 d, $n = 10$ sham and $n = 17$ SCI]. Insets in (**A–C**) are enlargements of the monosynaptic reflex while single arrows indicate the monosynaptic reflex and the stimulus artifact (St.). Bottom traces in (**C**) illustrate a dorsal root-evoked alternating locomotor-like activity recorded from opposite L5 ventral roots, with the corresponding cross-correlation histograms for left/right relationship. Parts of the recordings in J–L indicated by shaded areas are shown at a faster time scale on the right-hand side. (**D–F**) Average peristimulus time histogram (PSTH, bin width: 20 ms) of dorsal root evoked L5 ventral root responses collected from sham-operated (black) or SCI rats (red) 1 d (**D**), 2–3 d (**E**) or 4–5 d (**F**) post-SCI. (**G–I,M–O**) Group means quantification of: events per rat detected over time windows of 10–40 ms and 500–15,000 ms post-stimulus for SLR and LLR, respectively (**G,H**), threshold for evoking ventral root responses (**i**) and of spontaneous activites (**M–O**) at different time points post-SCI. *$p<0.05$, **$p<0.01$, ***$p<0.001$ comparing sham versus SCI groups; two-way ANOVA, Bonferroni's post-test. Data are means ± SD. Underlying numerical values can be found in the *Figure 3—source data 1*.

The online version of this article includes the following source data and figure supplement(s) for figure 3:

**Source data 1.** Values displayed in bar plots shown in *Figure 3G-I, M-O*.

**Figure supplement 1.** Stability of both hyperreflexia and spontaneous activities in spinal cord isolated from neonatal rats with SCI.

**Figure supplement 1—source data 1.** Values displayed in bar plots shown in *Figure 3—figure supplement 1B–D, F*.

## Calpain is upstream of the hyperexcitability of SCI motoneurons

To further study the role of calpain activity in spasticity, we examined the acute effects of MDL28170 on the spinal cord hyperexcitability in vitro (from 4 to 5 d post-SCI). Bath-applied MDL28170 (30 µM, 60 min) reduced dorsal root-evoked ventral root responses (*Figure 4A*) to the point that the PSTH approximated a unimodal distribution (*Figure 4B*). Both SLR and LLR decreased ($p<0.05$; *Figure 4C*) without affecting the monosynaptic reflex ($p>0.05$; insets in *Figure 4A* and histograms in *Figure 4C*). However, LLR remained higher compared to responses recorded in spinal cords from sham-operated rats ($p<0.01$). Likewise, spontaneous bursting activity became scarce, shorter and smaller in the presence of MDL28170 ($p<0.05$; *Figure 4D–E*) and did not differ from that recorded in sham animals ($p>0.05$). In adult rats, we previously demonstrated a causal relationship between calpain-mediated cleavage of Nav1.6 channels, up-regulation of $I_{NaP}$, and spasticity (*Brocard et al., 2016*). Consistent with these results, SCI in neonates enhanced a calpain-mediated cleavage of Nav channels, as pointed out by the increased density of a ~ 120 kDa band on membrane fraction of Western blots probed by a pan-Nav antibody (*Figure 4F*, *left lane*). This band was almost absent in intact animals (see Figure 4A in *Brocard et al., 2016*). Interestingly, the acute bath application of MDL28170 reduced the ~120 kDa fragment ($p<0.05$; *Figure 4F*, *right lane*) without affecting the prominent ~250 kDa representing the full-length form of Nav channels ($p>0.05$; *Figure 4F*). Consistent with, the MDL28170-induced decrease of the ~120 kDa fragment observed on membrane fraction was associated with a decrease in the density of immunostaining for Nav1.6, the main α-subunits expressed at the axon initial segment of lumbar motoneurons ($p<0.001$; *Figure 4G*). We performed voltage-clamp recordings in spinal cord slices to examine the functional impact of MDL28170 on biophysical properties of $I_{NaP}$ in lumbar SCI motoneurons. MDL28170 reduced by 16% the amplitude of $I_{NaP}$ ($p<0.05$; *Figure 4H*) without affecting its voltage activation threshold ($p>0.05$; *Figure 4H*). In contrast, MDL28170 did not change $I_{NaP}$ recorded from motoneurons in sham-operated controls ($p<0.05$; *Figure 4—figure supplement 1A–B*).

In addition to the up-regulation of $I_{NaP}$, down-regulation of KCC2 after SCI disinhibits motoneurons and leads to spasticity as well (*Boulenguez et al., 2010*). However, a causal relationship between calpain and the down-expression of KCC2 after SCI has never been explored. Bath-applied MDL 28170 increased by 21% the expression of KCC2 in its oligomeric ~240 kDa form ($p<0.05$; *Figure 4I*) without affecting the monomeric ~140 kDa form ($p>0.05$; *Figure 4I*). Furthermore, the addition of calpain-I in spinal cord homogenates from intact neonatal rat, dose-dependently reduced the expression of KCC2 ($p<0.01$; *Figure 4—figure supplement 2*). By immunohistochemical staining, we analyzed the expression of KCC2 on the pool of lumbar (L4-L5) motoneurons. MDL 28170 increased KCC2 staining in the lateral ventral horn where motoneurons are located ($p<0.05$; *Figure 4J*). We further studied the functional impact on $E_{IPSP}$ which was previously shown to be

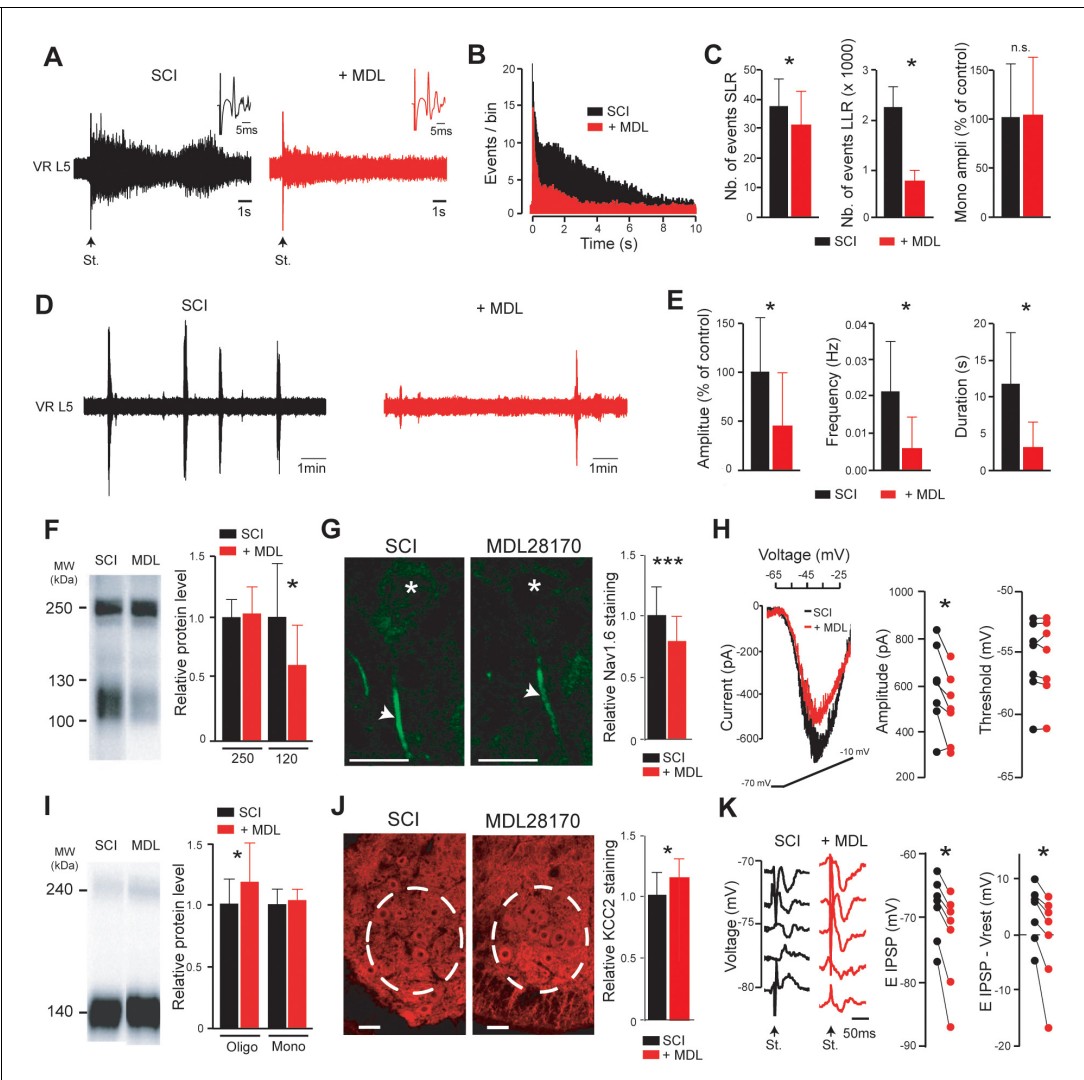

**Figure 4.** Inhibition of calpain restores the excitatory/inhibitory equilibrium of motoneurons. (A,D) Representative L5 ventral root activities (VR L5) evoked by a supramaximal stimulation (St.) of the ipsilateral dorsal root (A) or occurring spontaneously (D) in spinal cords isolated from SCI rats (4–5 d post-SCI; n = 6 rats) before (black) and after (red) bath-applying MDL28170 (30 µM, 60 min). Insets in (A) illustrate enlargements of the monosynaptic response. (B) Average peristimulus time histogram (PSTH, bin width: 20 ms) of dorsal root evoked L5 ventral root responses before (black) and after (red) adding MDL28170. (C,E) Group means quantification of: the monosynaptic reflex and events per rat detected over time windows of 10–40 ms and 500–15,000 ms post-stimulus for SLR and LLR, respectively (C) and of spontaneous activities (E). *p<0.05, comparing data collected before (black) and after MDL28170 (red); Wilcoxon paired test. (F,I) Pan-Nav (F) and KCC2 (I) immunoblots of lumbar segments from SCI rats (5 d post-SCI) bath-applied with vehicle (left lane) or MDL28170 (30 µM, right lane). On the right-hand side, quantification of immunoblots in MDL28170-treated samples (red, n = 12 rats for Pan-Nav, n = 8 rats for KCC2) normalized to vehicle-treated samples (black, n = 11 rats for Pan-Nav, n = 7 rats for KCC2). *p<0.05; comparing vehicle- to MDL28170-treated samples; Mann-Whitney test. (G,J) Representative single optical sections showing immunostaining in vehicle- (left) and MDL28170-treated (right) lumbar enlargement (L4–L5) from SCI rats (5 d post-SCI), against Nav1.6 α-subunit expressed in AISs of motoneurons (G) or KCC2 expressed in ventral horns (J). Asterisks in (G) indicate the motoneuron nucleus position and arrows their AISs. Dotted circles in (J) surround the ROI where KCC2 staining was quantified. Scale bars, 20 µm and 40 µm in (G) and (J), respectively. On the right-hand side, relative immunostaining intensities obtained with Nav 1.6 (G) and KCC2 (J) antibodies in MDL28170-treated samples (red) normalized to vehicle-treated controls (black). For Nav 1.6: N = 71 cells each from three rats per group; For KCC2: N = 47 slices each from three rats per group. *p<0.05, ***p<0.001 comparing vehicle-treated versus MDL28170-treated groups; t-test. (H) Representative leak-subtracted $I_{NaP}$ in a lumbar motoneuron (visualized as the largest cells located in layer IX of slices) from SCI rat (5 d post-SCI) and evoked by a slow (12 mV/s) voltage ramp increase from −70 mV to −10 mV over 5 s before (black) and after (red) bath-applying MDL28170 (30 µM, 30–60 min, n = 6 cells). Quantification of $I_{NaP}$ on the right-hand side. *p<0.05, Wilcoxon paired test. (K) IPSPs evoked by stimulation (St.) of the ventral funiculus of the spinal cord at different holding potentials in a lumbar motoneuron (identified by the antidromic response to stimulation of the ventral roots) from SCI rat (5 d post-SCI) before (black) and after (red) bath-applying MDL28170 (30 µM, 30–60 min, n = 7 cells). Quantification of $E_{IPSP}$ (left) and driving force ($E_{IPSP}$-$V_{rest}$, right) on the right-hand side. *p<0.05; Wilcoxon paired test (n = 7 cells). Data are mean ± SD. Underlying numerical values can be found in the *Figure 4—source data 1*.

*Figure 4 continued on next page*

*Figure 4 continued*

The online version of this article includes the following source data and figure supplement(s) for figure 4:

**Source data 1.** Values displayed in *Figure 4C, E-K*.
**Figure supplement 1.** Calpain inhibition affects neither $I_{NaP}$ nor $E_{IPSP}$ in lumbar motoneurons recorded from neonatal intact rats.
**Figure supplement 1—source data 1.** Values displayed in plots shown in *Figure 4—figure supplement 1B, D*.
**Figure supplement 2.** Cleavage of KCC2 channels by calpain-1.
**Figure supplement 2—source data 1.** Values displayed in bar plots shown in *Figure 4—figure supplement 2B, D*.

depolarized in lumbar motoneurons from neonatal rats with SCI (*Boulenguez et al., 2010*; *Bos et al., 2013*). Acute bath application of MDL28170 hyperpolarized motoneuronal $E_{IPSP}$ ($p<0.05$, *Figure 4K*). Because MDL28170 had no effect on the resting membrane potential ($V_{rest}$), the driving force on chloride ($E_{IPSP}$-$V_{rest}$) was significantly more negative ($p<0.05$, *Figure 4K*). MDL28170 had no effect on $E_{IPSP}$ of motoneurons recorded from sham-operated animals ($p<0.05$; *Figure 4—figure supplement 1C–D*).

## $I_{NaP}$-blockers or a KCC2-enhancer normalize the excitability of the spinal cord after SCI

We examined whether the SCI-induced spinal cord hyperexcitability in vitro can be reduced when persistent inward currents or KCC2 are targeted. Riluzole below 10 µM depresses $I_{NaP}$ in lumbar motoneurons without altering glutamatergic transmission (*Tazerart et al., 2008*). Bath-applied riluzole (5 µM) normalized the excitability of the SCI spinal cord. The SLR was decreased and the LLR was abolished without affecting the monosynaptic reflex ($p<0.05$, $p<0.01$ and $p>0.05$, respectively; *Figure 5A–C*). In addition, spontaneous bursting activity was markedly reduced ($p<0.01$; *Figure 5D–E*). Similar results were obtained for dorsal root-evoked ($p<0.05$; *Figure 5F–H*) or spontaneous bursting ($p<0.001$; *Figure 5I–J*) motor outputs when the Nav1.6-mediated current was specifically blocked by 4,9-anhydro-tetrodotoxin (200 nM; 4,9-ah-TTX (*Rosker et al., 2007*). Conversely, the selective pharmacological enhancement of $I_{NaP}$ with 100 nM of the steroidal alkaloid veratridine (*Alkadhi and Tian, 1996*; *Tazerart et al., 2008*) triggered LLR ($p<0.01$; *Figure 5—figure supplement 1A–C*) and increased spontaneous bursting activity ($p<0.05$; *Figure 5—figure supplement 1D–E*). The veratridine-induced motor responses were occluded by riluzole (5 µM, *Figure 5—figure supplement 1A–C*). Blockade of the second persistent inward current '$I_{CaP}$' by the L-type $Ca^{2+}$ channel blocker nifedipine (20 µM) did not reduce either hyperreflexia or spontaneous bursting activity ($p>0.05$; *Figure 5—figure supplement 2A–E*). However, the L-type $Ca^{2+}$ channel enhancer Bay-K (10 µM) triggered LLR and spontaneous bursting activity in isolated spinal cord from control intact neonatal rats ($p<0.05$; *Figure 5—figure supplement 2F–J*). The role of KCC2 in SCI-induced hyperexcitability was explored with the KCC2 activator, prochlorperazine (PCPz) (*Liabeuf et al., 2017*). PCPz used at the minimal concentration (10 µM) to restore KCC2 function after SCI (*Liabeuf et al., 2017*), reduced the LLR and spontaneous bursting activity in amplitude and frequency ($p<0.01$, $p<0.01$ and $p<0.05$ respectively; *Figure 5K–O*) with no effects on SLR or monosynaptic reflex amplitude.

## The SCI-induced alteration of $I_{NaP}$ and $E_{IPSP}$ promotes spinal hyperexcitability

The combination of up- and down-regulation of $I_{NaP}$ and $E_{IPSP}$ by calpain might be critically involved in spinal hyperexcitability. To study a possible synergistic effect between $I_{NaP}$ and $E_{IPSP}$ in increasing the excitability of the spinal cord after SCI, we pharmacologically mimicked the SCI-induced alteration of $I_{NaP}$ and $E_{IPSP}$ in spinal cords isolated from control intact neonatal rats. With a concentration of 60 nM, veratridine mimicked the $I_{NaP}$ increase caused by SCI (*Brocard et al., 2016*) (see also Figure 6 in *Tazerart et al., 2008*). A dose of 30 µM KCC2 antagonist, DIOA, can also reproduce the ~10 mV depolarizing shift of $E_{IPSP}$ reported in motoneurons after SCI (*Jean-Xavier et al., 2006*; see also Figure 4 in *Boulenguez et al., 2010*). At these concentrations, neither veratridine nor DIOA alone triggered either hyperreflexia or an increase in spontaneous bursts (gray, $p>0.05$; *Figure 6A–F*). However, when veratridine and DIOA were co-applied, enhanced LLR and spontaneous bursts were triggered (red, *Figure 6A–H*).

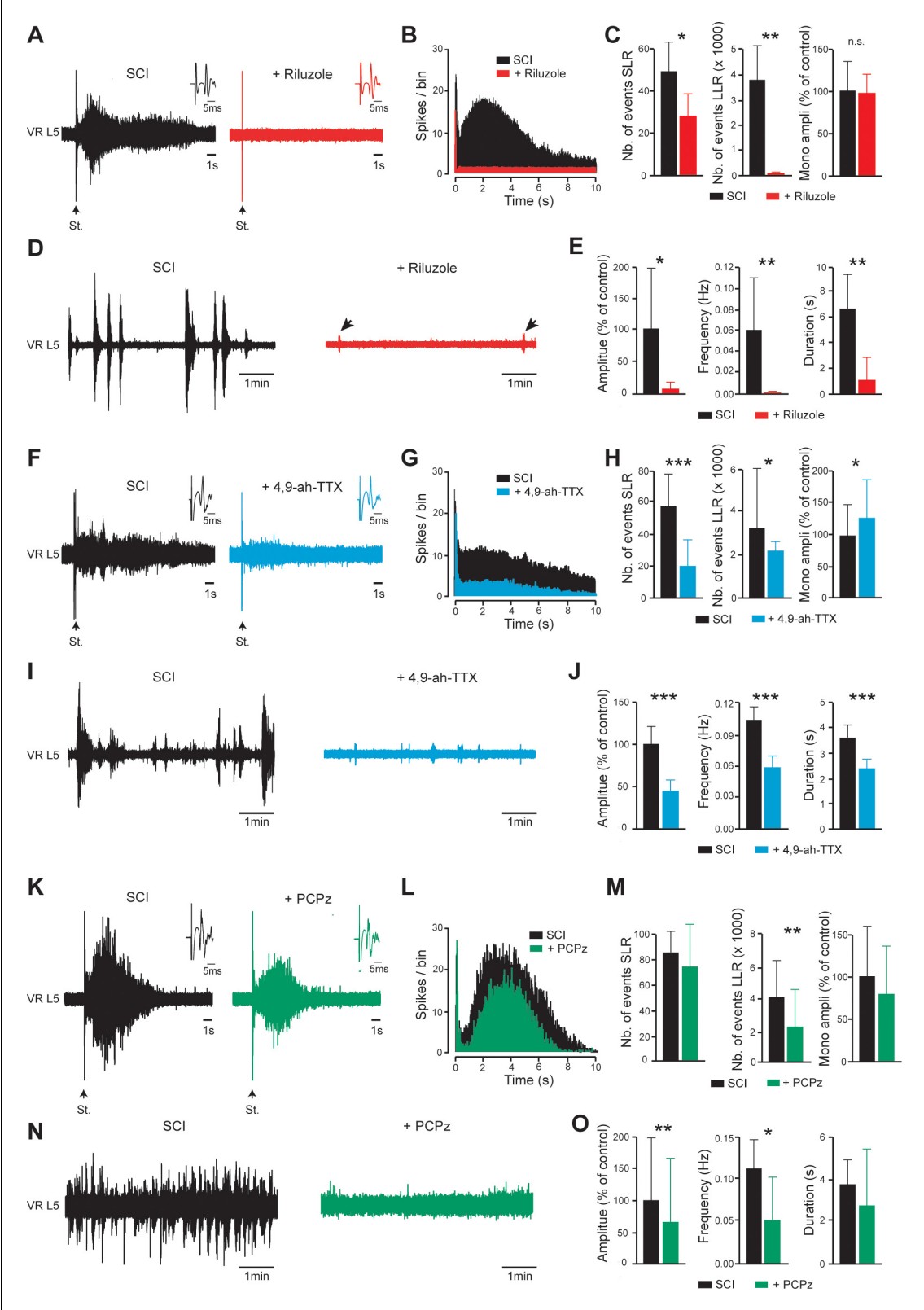

**Figure 5.** $I_{NaP}$-blockers or a KCC2-enhancer normalize the excitability of the spinal cord below SCI. (A,D,F,I,K,N) Representative L5 ventral root activities (VR L5) evoked by a supramaximal stimulation (St.) of the ipsilateral dorsal root (A,F,K) or occurring spontaneously (D,I,N) in spinal cords isolated from SCI rats (4–5 d post-SCI) before (black) and after bath-applying 5 µM riluzole (A,D); red, 30 min, *n* = 8 rats), 200 nM 4,9-ah-TTX (F,I; blue, 30 min, *n* = 15 rats for F, n = 20 rats for I) or 10 µM PCPz (K,N; green, 30 min, *n* = 9 rats for K, n = 11 rats for N). Insets in (A,F,K) illustrate enlargements

*Figure 5 continued on next page*

*Figure 5 continued*

of the monosynaptic response. (**B,G,L**) Average peristimulus time histogram (PSTH, bin width: 20 ms) of dorsal root evoked L5 ventral root responses before (black) and after (color) bath-applying the above-mentioned drugs. (**C,E,H,J,M,O**) Group means quantification of: the monosynaptic reflex and events per rat detected over time windows of 10–40 ms and 500–15,000 ms post-stimulus for SLR and LLR, respectively (**C,H,M**), and spontaneous activities (**E,J,O**). *p<0.05, **p<0.01, ***p<0.001, comparing data collected before and after adding drugs mentioned above; Wilcoxon paired test. Data are mean ± SD. Underlying numerical values can be found in the ***Figure 5—source data 1***.

The online version of this article includes the following source data and figure supplement(s) for figure 5:

**Source data 1.** Values displayed in bar plots shown in ***Figure 5C, E, H, J, M, O***.

**Figure supplement 1.** The $I_{NaP}$-enhancer veratridine used at 100 nM triggers riluzole-sensitive hyperreflexia and spontaneous activities in isolated spinal cords from intact neonatal rats.

**Figure supplement 1—source data 1.** Values displayed in bar plots shown in ***Figure 5—figure supplement 1C, E***.

**Figure supplement 2.** $I_{CaP}$ does not contribute to hyperexcitability caudal to SCI.

**Figure supplement 2—source data 1.** Values displayed in bar plots shown in ***Figure 5—figure supplement 2C, E, H, J***.

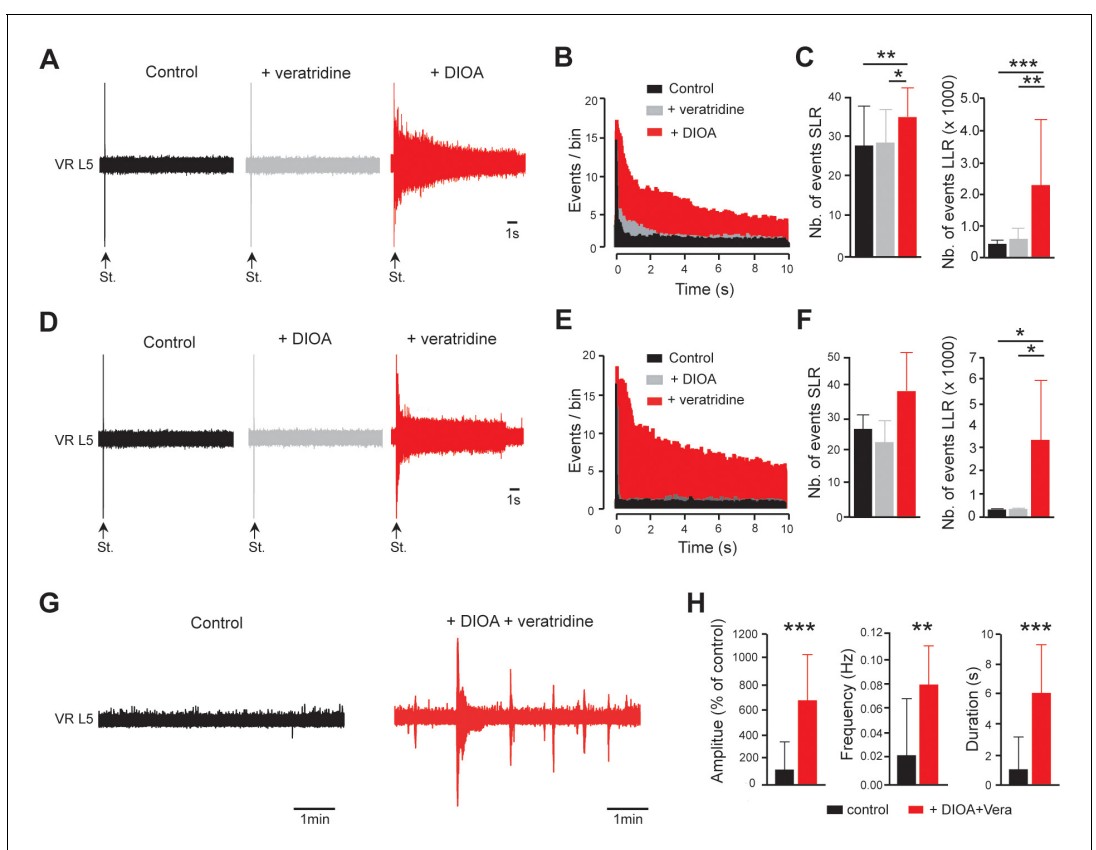

**Figure 6.** $E_{IPSP}$ and $I_{NaP}$ act synergistically to promote spinal hyperexcitability. (**A**) Representative L5 ventral root activities (VR L5) evoked by a supramaximal stimulation (St.) of the ipsilateral dorsal root in spinal cords isolated from intact rats (4–5 d old, n = 11 rats) before (black) and after (gray) the application of veratridine (60 nM, 30 min) followed by the superfusion (red) of DIOA (30 μM, 30 min). (**D**) Same experiments (n = 6 rats) as in (**A**) but by applying DIOA (gray) before veratridine (red). (**B,E**) Average peristimulus time histogram (PSTH, bin width: 20 ms) of dorsal root evoked L5 ventral root responses recorded either in the presence of veratridine or DIOA alone (gray), or in the presence of both (red). (**C,F**) Group means quantification of events per rat detected over time windows of 10–40 ms and 500–15,000 ms post-stimulus for SLR and LLR, respectively. *p<0.05, **p<0.01, ***p<0.001 repeated measures one-way ANOVA. (**G**) Representative spontaneous activities recorded on L5 ventral root (VR L5) in spinal cords isolated from intact rats (4–5 d old, n = 19 rats) before (black) and after (red) the superfusion of 60 nM veratridine with 30 μM DIOA (30 min). (**H**) Group means quantification of spontaneous activities. **p<0.01, ***p<0.001 Wilcoxon paired test. Data are mean ± SD. Underlying numerical values can be found in the ***Figure 6—source data 1***.

The online version of this article includes the following source data for figure 6:

**Source data 1.** Values displayed in bar plots shown in ***Figure 6C, F, H***.

# Discussion

Numerous studies on adult animal models delineate the hyperexcitable state of motoneurons as one of the causes of spasticity, presumably because of an increase of $I_{NaP}$ and a decrease of KCC2 expression. The present study performed in neonatal rats identifies the activation of calpains as the upstream mechanism of the hyperexcitability of motoneurons after SCI. Calpain-mediated cleavage of Nav channels and KCC2 up- and down-regulates $I_{NaP}$ and inhibition, respectively, in lumbar motoneurons. The reduction of calpain activity normalizes $I_{NaP}$ and the strength of inhibitory transmission, and thereby alleviates spasticity. Altogether, these findings uncover a new cellular mechanism contributing to spasticity and provide novel therapeutic targets. Rather than targeting Nav channels or KCC2 independently, inhibition of calpains appears as a promising therapy by targeting two pathways that are crucial for the development of spasticity, 'killing two birds with one stone'.

The main spasticity symptoms develop within one week following SCI in neonatal rats, and correlate with an early appearance of hyperexcitability in the spinal cord. The emergence of spasticity is surprisingly faster than that of adult animals, which appears several weeks after the injury. A more active calpain-mediated proteolysis during early development may explain this age-dependent discrepancy. Indeed, calpain proteolysis is necessary for the pruning of axons during development (*Yang et al., 2013*) and the natural calpain inhibitory peptide, calpastatin, increases slowly in the CNS during the first weeks of life (*Li et al., 2009*). The lower presence of this inhibitor in newborn rats may explain why hyperexcitability after a neonatal SCI developed faster. Also at birth, rat motoneurons overexpress NMDA receptors (*Vinay et al., 2000*) which have been demonstrated to be efficient in activating calpains in pathological conditions (*Zhou et al., 2012*). In sum, the low expression of calpastatin combined with a high expression of NMDA receptors may contribute to a fast increase of motoneuronal excitability after a SCI in newborn rats.

This hyperexcitability emerges in vitro in the form of high-frequency spontaneous bursting activity and enhanced, prolonged sensory-evoked LLR. We provided evidence that both are dependent on $I_{NaP}$ and KCC2 dysfunction. $I_{NaP}$ plays an important role in the operation of the spinal locomotor network (*Tazerart et al., 2007*; *Tazerart et al., 2008*; *Brocard et al., 2010*; *Bouhadfane et al., 2013*; *Brocard et al., 2013*) and its increase after SCI promotes prolonged plateau potential firing in motoneurons and hyperreflexia, both in animal models and humans (*Li and Bennett, 2003*; *Li et al., 2004*; *Kitzman, 2009*; *Theiss et al., 2011*). Supersensitivity of motoneurons to residual serotonin after SCI leads to the facilitation of $I_{NaP}$ (*Harvey et al., 2006a*; *Harvey et al., 2006b*). Here, we demonstrate that calpain-dependent cleavage of Nav channels also upregulates $I_{NaP}$ in lumbar motoneurons. If serotonin recruits calpains after SCI, similarly to what observed in *Aplysia* motoneurons (*Bougie et al., 2012*), remains to be tested.

The exact mechanisms involved in the facilitation of $I_{NaP}$ by calpain are not fully understood. As previously observed after a traumatic brain injury (*von Reyn et al., 2009*), proteolyzed Nav channels remain in the plasma membrane and thus might interfere with biophysical properties of full-length voltage-gated channels (*Michailidis et al., 2014*). On the other hand, some determinants governing the inactivation of Nav channels are sensitive to proteases, which can increase $I_{NaP}$ (*Armstrong et al., 1973*; *Gonoi and Hille, 1987*; *Clarkson, 1990*). Whatever the mechanisms, a clear relationship is given by our previous study in which calpain-mediated cleavage of Nav1.6 channels in HEK-293 cells up-regulates $I_{NaP}$ (*Brocard et al., 2016*). Because Nav1.6 channels are highly expressed in spinal motoneurons (*Alessandri-Haber et al., 2002*; *Duflocq et al., 2008*; *Brocard et al., 2016*), their cleavage likely contributes to increase $I_{NaP}$ after SCI. This is consistent with our result that pharmacological inhibition of Nav1.6 channels significantly reduced hyperexcitability after SCI. Further support comes from mutation of Nav1.6 channels that leads to severe myoclonic spasms with a fivefold increase of $I_{NaP}$ (*Veeramah et al., 2012*). It is noteworthy that the rescue of $I_{NaP}$ by MDL28170 is only partial. Although an incomplete block of calpains by MDL28170 cannot be excluded, calpain-independent mechanism(s) may contribute to increase $I_{NaP}$. Since the activation of $5HT_2$ receptors facilitates $I_{NaP}$ (*Harvey et al., 2006b*), the constitutive activation of $5HT_2$ receptors after SCI (*Murray et al., 2010*) might be one of them. Also, the amount of full-length Nav channels (~250 kDa) at the plasma membrane never changed after either the post-SCI activation of calpains (*Brocard et al., 2016*) or their inhibition by a MDL28170 treatment (see *Figure 3F*). As a compensation mechanism depending on the number of proteolytic fragments, it is possible that the

translocation of fragments into the nucleus maintains constant the expression level of the full-length channel at the membrane by regulating the Nav channel transcription (*Onwuli et al., 2017*).

Synaptic disinhibition of motoneurons after SCI also contributes to spasticity and results from a decreased expression of KCC2 (*Boulenguez et al., 2010*). Although a down-regulation of KCC2 has been described after SCI (*Boulenguez et al., 2010*; *Côté et al., 2014*; *Mòdol et al., 2014*; *Chang et al., 2018*; *Chen et al., 2018*), the causal molecular mechanisms remain unknown. Here, we provide evidence that activation of calpains is one of the molecular basis for disinhibition of motoneurons after SCI. A direct relationship between calpain and KCC2 is given by our studies on spinal cord homogenates, in which exogenous application of active calpains reduces KCC2 in its oligo and monomeric forms. The sensitivity of KCC2 to calpain is consistent with the presence of two predicted sites (PEST motifs) for cleavage by calpain within the C-terminal domain (*Mercado et al., 2006*; *Acton et al., 2012*). Since the ability of KCC2 to extrude Cl⁻ requires the C-terminal domain (*Mercado et al., 2006*; *Acton et al., 2012*) and that the KCC2 antibody targets the C-tail, its cleavage by calpains after SCI likely accounts for both the immunostaining decrease of KCC2 at the plasma membrane and the loss of chloride extrusion resulting in the depolarizing shift of $E_{IPSP}$ in motoneurons (*Boulenguez et al., 2010*). The tight relationship between the calpain-mediated proteolysis of KCC2 and the altered Cl⁻ homeostasis is supported by our observation that an acute inhibition of calpains by MDL28170 rapidly restores both $E_{IPSP}$ and KCC2. The rapid rescue of KCC2 by the acute application of MDL28170 is in agreement with the fast pharmacodynamic of MDL28170 to cross the brain blood barrier (*Markgraf et al., 1998*) and the extremely high rate (minutes) in KCC2 turnover at the plasma membrane (*Lee et al., 2007*). Regarding the rescue of KCC2 by MDL28170, oligomers appear more sensitive to calpains than monomers after SCI. As phosphorylation regulates the vulnerability of substrates to calpains (*Sprague et al., 2008*), the post-SCI dephosphorylation of KCC2 at the serine 960 (*Mòdol et al., 2014*), in the vicinity of PEST motifs, might contribute to sensitize more oligomers to calpains.

Our results suggest that KCC2 and $I_{NaP}$ appear to cooperate in promoting spinal hyperexcitability after SCI. In line with this, a modeling investigation demonstrated that the depolarizing shift of $E_{IPSP}$ after SCI facilitates the recruitment of persistent inward currents in motoneurons (*Venugopal et al., 2011*). The depolarizing shift of $E_{IPSP}$ after a decrease in KCC2 may enable the unusual long-lasting depolarization reported in motoneurons after SCI (*Li et al., 2004*), which is required to recruit $I_{NaP}$. In turn, $I_{NaP}$ will promote plateau potentials resulting in self-sustaining spiking in motoneurons (*Li and Bennett, 2003*; *Bouhadfane et al., 2013*) that drives spasticity (*Bennett et al., 2001*; *Gorassini, 2004*; *Li et al., 2004*). Furthermore, termination of plateau potentials by hyperpolarizing the motoneuron (*Hounsgaard et al., 1984*) will be much more difficult in a context of disinhibition; this may explain the fasciculation-like contractions commonly recorded in spastic subjects (*Gorassini, 2004*; *Winslow et al., 2009*; *Zijdewind and Thomas, 2012*). Therefore, $I_{NaP}$-blockers such as riluzole, or KCC2 enhancers likely alleviate spasticity (*Brocard et al., 2016*; *Liabeuf et al., 2017*) by decoupling the tandem response driven by enhanced $I_{NaP}$ and decreased KCC2. The riluzole is currently approved for humans affected by Amyotrophic Lateral Sclerosis (ALS). Our data provide strong preclinical evidence for translation to chronic SCI subjects, a process that will likely be facilitated by clinical trials that are currently in progress to test the neuroprotective effects of riluzole in the acute phase of SCI (*Grossman et al., 2014*).

We identify calpain as the upstream mechanism responsible for the hyperexcitability of motoneurons after SCI. Calpains exist in the CNS mainly as two major isoforms, μ-calpain (calpain-I) and m-calpain (calpain-II) that differ on the range of [$Ca^{2+}$] required for their activation (μM and mM, respectively). As previously observed in adult rodents (*Springer et al., 1997*; *Du et al., 1999*; *Schumacher et al., 1999*; *Yu et al., 2013*), we found that the expression of calpain-I increases after SCI in neonatal rats. However, the protease protein expression does not necessarily correspond to enzyme's catalytic activity. Furthermore, calpain-I and -II may have opposite functions. After a traumatic brain injury (TBI), calpain-I and -II appear neuroprotective and neurodegenerative, respectively (*Baudry and Bi, 2016*). Therefore, the respective contribution of calpain-I and -II in the hyperexcitability of motoneurons after SCI remains to be clarified.

Calpain-mediated changes in motoneurons alone are unlikely to account for all the post-injury spinal hyperexcitability. Premotor excitatory interneurons, including locomotor network-related interneurons, may play a critical role in initiating muscle spasm activity (*Husch et al., 2012*; *Bellardita et al., 2017*; *Lin et al., 2019*). In line with this, we found that most sensory-evoked LLRs

are composed of a fictive locomotor episode, suggesting enhanced sensory recruitment of the locomotor central pattern generator (CPG). This leads us to believe that part of muscle spasm, and especially clonus (involuntary rhythmic contractions), might be a manifestation of a hyperexcitable locomotor CPG. Enhanced sensory-evoked LLR has been previously described in the sacrocaudal spinal cord from adult SCI rodents but differs in some aspects from our results with neonatal lumbar motoneurons. Indeed, $I_{CaP}$ contributes to sacrocaudal hyperreflexia by promoting plateaus in motoneurons (*Li and Bennett, 2003*; *Li et al., 2004*) while disinhibition appears negligible (*Bellardita et al., 2017*). The $I_{CaP}$ appears to have a negligible role in lumbar hyperreflexia from neonatal SCI rats, in line with previous work showing a full maturation of L-type $Ca^{2+}$ channels at second/third post-natal weeks (*Jiang et al., 1999*). However, we also demonstrate that some of these channels are already expressed in the first post-natal week, as their pharmacological activation by Bay-K triggered LLRs. Thus, it is likely that the contribution of L-type $Ca^{2+}$ channels to the LLR increases with age. Alternatively, temperature and ionic composition of the extracellular medium may account for the relative contribution of $I_{NaP}$ and $I_{CaP}$ in generating LLRs. In vitro studies in adults were performed at room temperature with high extracellular $[Ca^{2+}]$ ($\geq 2.5$ mM) (*Li and Bennett, 2003*; *Li et al., 2004*; *Bellardita et al., 2017*), far from physiological conditions [body temperature 37°C; 1.2 mM of $Ca^{2+}$ in the CSF (*Nicholson et al., 1977*; *Jones and Keep, 1988*; *Brocard et al., 2013*). As a consequence, $Ca^{2+}$ currents are potentiated (*Carlin et al., 2000*; *Carlin et al., 2009*), $I_{NaP}$ is reduced (*Tazerart et al., 2008*) and thermosensitive $I_{NaP}$-dependent plateaus in motoneurons are dampened down (*Bouhadfane et al., 2013*). Also, the disinhibition seen in the SCI lumbar enlargement, linked to the decrease of KCC2 (*Boulenguez et al., 2010*), seems unimportant in the sacrocaudal spinal cord (*Bellardita et al., 2017*). Diversity of KCC2 expression between motoneurons innervating distinct muscles (*Ueno et al., 2002*) or variation in the sacral inhibitory circuitry (*Jankowska et al., 1978*) may account for this discrepancy. Spinal cord level-specific differences in neuronal responses to SCI are probable with respect to the special anatomy and function of the sacrocaudal spinal cord (*Ritz et al., 1992*; *Ritz et al., 2001*).

To conclude, our study sheds light on the etiology of spasticity and opens novel perspectives to develop therapies by targeting calpains. The discovery of calpain as a new upstream mechanism leading to spasticity, is of special importance given the lack of translational results obtained from previously tested therapeutic approaches. Since spinal maladaptive mechanisms triggered by calpain start taking place within the first hours after injury in our SCI model, it is conceivable that early therapies against calpain might show a higher effectiveness. Because altered chloride homeostasis and proteolysis of Nav channels have been implicated in other neurological disorders such as traumatic brain injury for which calpains are recruited (*Kahle et al., 2008*; *von Reyn et al., 2009*; *Wang et al., 2018*), the involvement of calpains might be broadened to other pathologies leading to an excitatory/inhibitory imbalance.

# Materials and methods

## Key resources table

| Reagent type (species) or resource | Designation | Source or reference | Identifiers | Additional information |
|---|---|---|---|---|
| Antibody | mouse monoclonal anti-PanNav | Sigma | clone K58/35 CAT#S8809 RRID:AB_477552 | (1:500) |
| Antibody | rabbit polyclonal anti-Na$_v$1.6 | Alomone | Cat # ASC-009 RRID:AB_2040202 | (1:200) |
| Antibody | rabbit polyclonal anti-KCC2 | Millipore | CAT# 07–432 RRID:AB_310611 | (1:400, 1:500) |
| Antibody | rabbit polyclonal anti-calpain-I | Ozyme | CAT#2556S | (1:500) |
| Antibody | rabbit polyclonal anti-calpain-II | Millipore | CAT#AB81023 RRID:AB_1586917 | (1:500) |

*Continued on next page*

*Continued*

| Reagent type (species) or resource | Designation | Source or reference | Identifiers | Additional information |
|---|---|---|---|---|
| Antibody | ImmunoPure goat HRP-conjugated mouse-specific antibody | Thermo Fisher Scientific | CAT#0031430 RRID:AB_228307 | (1:40000) |
| Antibody | AlexaFluor-488 goat anti-mouse IgG2b | Thermo Fisher Scientific | CAT# A-21141 RRID:AB_141626 | (1:800) |
| Antibody | AlexaFluor-546 F(ab')two goat anti-rabbit IgG | Thermo Fisher Scientific | CAT# A-11071 RRID:AB_2534115 | (1:400) |
| Chemical compound, drug | NaCl | Sigma-Aldrich | CAT# 71376 | |
| Chemical compound, drug | KCl | Sigma-Aldrich | CAT# P3911 | |
| Chemical compound, drug | $NaH_2PO_4$ | Sigma-Aldrich | CAT# S0751 | |
| Chemical compound, drug | $MgSO_4$ | Sigma-Aldrich | CAT# 1880 | |
| Chemical compound, drug | $CaCl_2$ | Sigma-Aldrich | CAT# 21115 | |
| Chemical compound, drug | $NaHCO_3$ | Sigma-Aldrich | CAT# S6014 | |
| Chemical compound, drug | D-glucose | Sigma-Aldrich | CAT# G8270 | |
| Chemical compound, drug | $K^+$-gluconate | Sigma-Aldrich | CAT# P1847 | |
| Chemical compound, drug | $MgCl_2$ | Sigma-Aldrich | CAT# M8266 | |
| Chemical compound, drug | HEPES | Sigma-Aldrich | CAT# H3375 | |
| Chemical compound, drug | EGTA | Sigma-Aldrich | CAT# E3889 | |
| Chemical compound, drug | ATP | Sigma-Aldrich | CAT# A9062 | |
| Chemical compound, drug | GTP | Sigma-Aldrich | CAT# G9002 | |
| Chemical compound, drug | Sucrose | Sigma-Aldrich | CAT# S9378 | |
| Chemical compound, drug | Cadmium Chloride | Sigma-Aldrich | CAT# 202908 | |

*Continued on next page*

*Continued*

| Reagent type (species) or resource | Designation | Source or reference | Identifiers | Additional information |
|---|---|---|---|---|
| Chemical compound, drug | Tetraethylammonium chloride | Sigma-Aldrich | CAT# 86616 | |
| Chemical compound, drug | Nifedipine | Sigma-Aldrich | CAT# N7634 | |
| Chemical compound, drug | Dimethylsulphoxide | Sigma-Aldrich | CAT# D8418 | |
| Chemical compound, drug | Paraformaldehyde | EMS | CAT# 15714 s | |
| Chemical compound, drug | Phosphate Buffered Saline | Argene Biomérieux | CAT# 33–011 | |
| Chemical compound, drug | Tissue-Tek OCT compound | VWR | CAT# 25608–930 | |
| Chemical compound, drug | Tris-buffered saline | Bio-world | CAT# 1053 00272 | |
| Chemical compound, drug | Triton X-100 | Sigma-Aldrich | CAT# T9284 | |
| Chemical compound, drug | CompleteMini | Roche diagnostic Basel | CAT#1183 6170001 | |
| Chemical compound, drug | Iodoacetamide | Sigma-Aldrich | CAT#I1149 | |
| Chemical compound, drug | Igepal CA-630 | Sigma-Aldrich | CAT#I8896 | |
| Chemical compound, drug | SDS | Sigma-Aldrich | CAT#05030–1 L-F | |
| chemical compound, drug | Acryl/Bisacrylamide solution | Biorad | CAT# 161–0146 | |
| Chemical compound, drug | PMSF | Sigma-Aldrich | CAT#93482 | |
| Chemical compound, drug | Pepstatin A | Sigma-Aldrich | CAT#77170 | |
| Chemical compound, drug | MDL28170 | Calbiochem | CAT#208722 | |
| Chemical compound, drug | Calpain 1 | Calbiochem | CAT#208712 | |
| Chemical compound, drug | Bay K8644 | Sigma-Aldrich | CAT# B112 | |

*Continued on next page*

*Continued*

| Reagent type (species) or resource | Designation | Source or reference | Identifiers | Additional information |
|---|---|---|---|---|
| chemical compound, drug | CNQX | Sigma-Aldrich | CAT# C127 | |
| Chemical compound, drug | DIOA | Sigma-Aldrich | CAT# D129 | |
| Chemical compound, drug | Veratridine | Sigma-Aldrich | CAT# V5754 | |
| Chemical compound, drug | AP5 | Tocris | CAT# 3693/10 | |
| Chemical compound, drug | PCPz | Tocris | CAT# 3287/100 | |
| Chemical compound, drug | Riluzole | Tocris | CAT# 0768/25 | |
| Chemical compound, drug | 4,9-ah-TTX | Focus Biomolecules | CAT# 10–3700 | |
| Strain, strain background (Rattus norvegicus) | Wistar rats | Charles River Laboratories | RRID: RGD_2308816 | |
| Software, algorithm | pClamp v10.3 | Molecular Devices | RRID: SCR_011323 | |
| Software, algorithm | ImageJ v1.50i | https://imagej.nih.gov/ij/ | RRID: SCR_003070 | |
| Software, algorithm | FluoView v5.0 | Olympus | RRID: SCR_014215 | |
| Software, algorithm | Image Lab v5.1 | Bio-Rad | RRID: SCR_014210 | |
| Software, algorithm | Graphpad Prism | Prism | RRID: SCR_002798 | |
| Other | Vibrating microtome | Leica | VT1000S RRID: SCR_016495 | |
| Other | Temperature controller | Warner Instruments | CL-100 | |
| Other | Nikon Eclipse microscope | Nikon | E600FN | |
| Other | Confocal microscope | Zeiss | LSM510 | |
| Other | Infrared-sensitive CCD camera | Hitachi | KP-200/201 | |
| Other | Digidata 1440a interface | Molecular Devices | N/A | |
| Other | Multiclamp 700B amplifier | Molecular Devices | N/A | |
| Other | Borosilicate glass capillaries | World Precision Instruments | CAT# TW150-4 | |
| Other | Sutter P-97 puller | Sutter Instruments | RRID: SCR_016842 | |

*Continued on next page*

*Continued*

| Reagent type (species) or resource | Designation | Source or reference | Identifiers | Additional information |
|---|---|---|---|---|
| Other | Knittel Glass coverslips | Dutscher | CAT# 900529 | |
| Other | Polysine slides | Thermoscientific | CAT# P4981 | |
| Other | Small Animal Ventilator | CWE | SAR-830/AP | |
| Other | Cardiotachometer | CWE | CT-1000 | |
| Other | Chemidoc imaging system XRS+ | Bio-rad | RRID: SCR_014210 | |

## Ethics statement

We made all efforts to minimize animal suffering and the number of animals used. All animal care and use conformed to the French regulations (Décret 2010–118) and were approved by the local ethics committee (Comité d'Ethique en Neurosciences INT-Marseille, CE Nb A1301404, authorization Nb 2018110819197361).

## Surgery and postoperative care

The spinal cord trans-section was performed in neonatal rats within the first 12 hr after birth. Animals were anesthetized by hypothermia. After a midline skin incision, a laminectomy was performed to expose lower thoracic segments of the spinal cord. The dura was opened and the spinal cord was completely transected at the T8-T9 segmental level. The lesion cavity was filled with sterile absorbable local hemostat Surgicoll. Finally, the wound was covered with Steri-Strips (3M Health Care, St. Paul, MN) and animals were kept warm and wet for 2 hr in cotton-wool swab impregnated with their mother smell before they returned to their home cage with their mother. The antibiotic amoxycilin (150 mg/kg, s.c.) was subcutaneously applied at the incision site just before suturing the skin to prevent bacterial infections. Note that only one-shot topical amoxycilin was administered to limit the potential emergence of antibiotic-resistant bacterial strains. Sham animals were submitted to all procedures except the spinal cord transection.

## Assessment of spastic motor behaviors

Animals were tested 4–5 days after SCI, when signs of spastic motor behaviors were visible such as excessive involuntary twitch/movement, exaggeration of reflexes... *For behavioral assessment* of spastic behaviors pups were removed from their home cage, weighted, and placed on a heating pad thermo-controlled at ~34.5 ± 1˚C. Dorsal view of the animals were recorded with a digital video camera. Recordings began 10 min after pups had been placed on the heating pad so as to ensure that pups were thermally stable. Then, a continuous 10 min recording was acquired after which time pups were returned to their home cages. The number of myoclonic twitching of the hindlimbs and the tail was scored in a single pass through the video record. *For the electrophysiological assessment* of spasticity, a stainless steel needle electrode was inserted transcutaneously into the triceps surae muscles (ankle extensors), and the reference electrode was placed subcutaneously on the back. Animals were slightly anesthetized by hypothermia before inserting electrodes. After a 20 min acclimation period, motor responses to pinch tail between the thumb and the index finger were recorded. EMG signals were amplified (100x) and bandpass filtered (300 Hz to 5 kHz; A-M Systems Amplifier, Everett, WA; model 1700) before sampling at 13.5 kHz (Digidata 1440A, Molecular Devices). The pressure manually applied to the last third of the tail was of increasing intensity until the appearance of a motor response in hindlimbs. The pressure (weight in grams) was recorded by a miniature pressure sensor placed between the thumb and the tail and monitored on line. Experimenters were not blinded during the whole procedure, as signs of spasticity were evident in SCI pups. However, experimenters were blind for the data analysis of electrophysiological experiments.

## In vitro preparations

Details of the in vitro preparations have been previously described (*Brocard et al., 2003*) and are only summarized here. *For the whole spinal cord preparation,* the spinal cord and ventral roots were removed from sacral segments up to $T_8$–$T_{10}$. The spinal cord was pinned down, ventral side up, in the recording chamber. All dissection and recording procedures were performed under continuous perfusion with aCSF composed of (in mM): 120 NaCl, 4 KCl, 1.25 $NaH_2PO_4$, 1.3 $MgSO_4$, 1.2 $CaCl_2$, 25 $NaHCO_3$, 20 D-glucose, pH 7.4 (32–34˚C). *For the slice preparation,* the lumbar spinal cords was isolated in ice-cold (<4˚C) artificial cerebrospinal fluid (aCSF) solution with the following composition (in mM): 232 sucrose, 3 KCl, 1.25 $KH_2PO_4$, 4 $MgSO_4$, 0.2 $CaCl_2$, 26 $NaHCO_3$, 25 D-glucose, pH 7.4. The lumbar spinal cord was then introduced into a 1% agar solution, quickly cooled, mounted in a vibrating microtome (Leica VT1000S) and sliced (350 µm) through the L4-5 lumbar segments. Slices were immediately transferred into the holding chamber filled with aCSF composed of (in mM): 120 NaCl, 3 KCl, 1.25 $NaH_2PO_4$, 1.3 $MgSO_4$, 1.2 $CaCl_2$, 25 $NaHCO_3$, 20 D-glucose, pH 7.4 (32–34˚C). Following a 1 hr resting period, individual slices were transferred to a recording chamber that was continuously perfused with the same medium heated to ~32˚C. All solutions were oxygenated with 95% $O_2$/5% $CO_2$.

## In vitro recordings and stimulation

*For the whole spinal cord preparation*, motor outputs were recorded using extracellular stainless steel electrodes placed in contact with right and left lumbar ventral roots (L5) and insulated with Vaseline. The ventral root recordings were amplified (×2,000), high-pass filtered at 70 Hz, low-pass filtered at 3 kHz, and sampled at 10 kHz. Custom-built amplifiers enabled simultaneous online rectification and integration (100 ms time constant) of raw signals. Monopolar stainless steel electrodes were also placed in contact with the dorsal roots and insulated with Vaseline to deliver a brief supramaximal stimulation (0.2 ms duration). Glass suction electrodes were sometimes used to stimulate the ventral funiculus at the $L_2$–$L_3$ level. After the pia had been removed, lumbar motoneurons were recorded intracellularly using glass microelectrodes filled with 2 M K-acetate (90–150 MΩ resistance). Intracellular potentials were recorded in the discontinuous current-clamp (DCC) mode (Axoclamp 2B amplifier; Digidata 1200 interface). Only neurons exhibiting a stable (>15 min) resting membrane potential were considered for analysis. Motoneurons were identified by the antidromic response to stimulation of the ventral roots. Stimulation of the ventral funiculus usually induced inhibitory postsynaptic potentials (IPSPs) in the presence of 2-amino-5-phosphonovaleric acid (AP5, 30–100 µM) and 6-cyano-7-nitroquin-oxaline-2,3-dione (CNQX, 3–10 µM). IPSPs were recorded at various holding potentials (500 ms-long current pulses). *For the slice preparation,* whole-cell patch-clamp recordings were performed in voltage-clamp mode from motoneurons located in the lateral ventral horn using a Multiclamp 700B amplifier (Molecular Devices). Motoneurons were visually identified with video microscopy (Nikon Eclipse E600FN) coupled to infrared differential interference contrast, as the largest cells located in layer IX. Only neurons with a membrane capacitance higher than 100 pF were considered. The image was enhanced with a Hitachi KP-200/201 infrared-sensitive CCD camera and displayed on a video monitor. Patch electrodes (2–4 MΩ) were pulled from borosilicate glass capillaries (1.5 mm OD, 1.12 mm ID; World Precision Instruments) on a Sutter P-97 puller (Sutter Instruments Company) and filled with intracellular solution containing (in mM): 140 $K^+$-gluconate, 5 NaCl, 2 $MgCl_2$, 10 HEPES, 0.5 EGTA, 2 ATP, 0.4 GTP, pH 7.3 (280 to 290 mOsm). Pipette and neuronal capacitive currents were canceled and, after breakthrough, the series resistance was compensated and monitored. Recordings were digitized on-line and filtered at 10 kHz (Digidata 1322A, Molecular Devices). The main characterization of $I_{NaP}$ was accomplished by slow ramp increase from −70 mV to −10 mV, slow enough (12 mV/s) to prevent transient sodium channel opening.

## Immunohistochemistry

Spinal cords were immersion-fixed for 1 hr in 0.25% PFA, washed in PBS and cryoprotected overnight at 4˚C in 20% sucrose in PBS. Lumbar spinal cords (L4-L5) were then frozen in OCT medium (Tissue Tec), cryosectioned (20 µm) and processed for immunohistochemistry. Sections from the control vs. SCI rats were mounted on the same slides and processed simultaneously. Slices were then (i) rehydrated in PBS at room temperature (15 min), (ii) permeated with 1% Bovin Serum Albumin

(BSA), 2% Natural Goat Serum (NGS) and 0.2% Triton x-100 (1 hr), (iii) incubated overnight at 4°C in the following affinity-purified rabbit $Na_v1.6$ (residues 1042–1061; 1:200; ASC009, Alomone) specific polyclonal antibodies, KCC2 (residues 932–1043; 1:400, 07–432, Millipore) (iv) washed in PBS (3 × 5 min), (v) incubated with fluorescent-conjugated secondary antibodies [Alexa 488- or 546-conjugated rabbit-specific antibodies (1:800 and 1:400; Lifetechnologies Carlsbad CA USA) used for visualization of the rabbit polyclonal antibodies] in a solution containing 1% BSA and 2% NGS (1.5 hr), (vi) washed in PBS 3 × 5 min, (vii) coverslipped with a gelatinous aqueous medium. In control experiments, the primary antiserum was either omitted or replaced with rabbit immunoglobulin fraction during the staining protocol. Sections were scanned using a laser scanning confocal microscope (Zeiss LSM510) in stacks of 1µm-thick optical sections at ×20 magnification and processed with the Fluoview software. Each optical section resulted from two scanning averages. We used identical settings, finely tuned to avoid saturation, for the whole series. Each figure corresponds to a projection image from a stack of optical sections.

## Membrane protein isolation and western blots

Tissues were collected from spinal cord lumbar enlargements and frozen after removing the dorsal and ventral roots. For the membrane fraction, corresponding to the plasma membrane-enriched fraction, samples were homogenized in ice-cold lysis buffer (320 mM sucrose, 5 mM Tris-HCL pH 7.5, 10 µM iodoacetamide) supplemented with protease inhibitors (CompleteMini, Roche diagnostic Basel, Switzerland). Unsolubilized material was pelleted by centrifugation at 7,000 g for 5 min. The supernatant was subjected to an additional centrifugation step at 18000 g for 70 min at 4°C. Pellets were collected and homogenized in ice cold lysis buffer (1% Igepal CA-630, Phosphate Buffer Saline 1X, 0.1% SDS, 10 µM iodoacetamide), supplemented with protease inhibitors (CompleteMini, Roche diagnostic). Protein concentrations were determined using a detergent-compatible protein assay (Bio-Rad, Hercules, CA, USA). Equal protein amounts (60 µg) from samples were size fractionated by 6% (vol/vol) SDS/PAGE from 40% Acryl/Bisacrylamide (29/1) commercial solution, transferred to a PVDF membrane and probed with either a mouse PanNav antibody (preserved sequence of the sodium channel α-subunit III-IV loop; 1:500; clone K58/35, Sigma), a polyclonal rabbit KCC2 antibody (1:500, 07–432, Millipore), a polyclonal rabbit calpain-I antibody (1:500, 2556S, Ozyme) or a polyclonal rabbit calpain-II antibody (1:500, AB81023, Millipore) at 4°C overnight in Tris-buffered saline containing 5% fat-free milk powder. The blot was then incubated for 1 hr at 22°C with an Immuno-Pure goat HRP-conjugated mouse-specific antibody (1:40,000 in blocking solution; Thermo Scientific, Waltham, MA, USA).

## Calpain cleavage assay

Lumbar spinal cords were homogenized in 400 µl sucrose buffer for membrane preparation (300 mM sucrose, 10 mM Tris base, 2 mM EDTA, 0.5 mM PMSF, and 1 µM pepstatin A) and centrifuged at 7,000 g for 5 min at 4°C. The supernatant was subjected to an additional centrifugation step at 18000 g, for 70 min at 4°C, and the pellet was resuspended in the same buffer. Protein concentrations were analyzed using the Bio-Rad Dc Protein Assay. Membrane preparations were then pretreated with or without the calpain inhibitor (30 µM MDL28170, Calbiochem) for 15 min on ice. Calpain 1 (0.5 to 3 U, Calbiochem) were added for 15 min at room temperature (24°C). The reaction was stopped after addition of the electrophoresis sample buffer.

## Data analysis

 *Softwares:* The Clampfit 10.7 software (Molecular devices) was used for analyzing electrophysiological data. The FluoView Sofware (version 5, Olympus) was used for quantifying immunostaining intensities. *Assessment of spastic motor behaviors:* For quantifying the number of twitches, the observer pressed the key of an event recorder when a twitching movement of any of the limbs and/or tail was detected. Myoclonic twitching was defined as a phasic uncontrolled movement of any part of the focal limbs and/or tail (*Gramsbergen et al., 1970*). Simultaneous twitching movements of any hinlimb(s) with the tail were scored as a single twitch. Observers were careful to distinguish an active movement (e.g., kicking, twisting, and pushing). *Extracellular recordings:* EMG recordings from the triceps surae muscles were quantified by measuring the duration of the response from rectified and low pass filtered (8 Hz) digital signals. The duration was determined as the time at which the

envelope of the signal exceeds a threshold. The threshold was set as the mean plus three times the standard deviation of the envelope during a period of inactivity. Threshold for mechanical stimuli was determined as the minimal pressure (weight in grams) applied to the tail for the appearance of a EMG response. At least five EMG recordings were analyzed from each animal. In response to a brief dorsal root stimulation, motor outputs recorded on the ventral roots from in vitro isolated spinal cord were quantified by cumulative counts of spikes generated in peristimulus time histograms (PSTHs; bins width: 20 ms) and calculated over a time window of 15,000 ms post-stimulation using a voltage peak detector. PSTHs were constructed from five consecutive rectified responses. We computed the transient short latency (SLR) and long-lasting reflexes (LLR) over time windows of 10–40 ms and 500–15,000 ms post-stimulus, respectively. Time windows were discretized into 20 ms-bins. For each bin, we calculated the number of events using a threshold search on ClampFit, the result was added in the corresponding time window. For each animal, we obtained one value for the SLR and one for the LLR, that were used for statistical analysis. Counts were corrected for spontaneous activity by subtracting the number of spontaneous events arising prior to the stimulus. To characterize spontaneous activities, the ventral root extracellular data were rectified integrated and smoothed with a time constant of 0.1 s. Amplitude, duration and frequency of spontaneous ventral root activities were measured with threshold-based event detection which determines the peak, the onset and end of bursts of activity. When oscillatory activities superimposed spontaneous bursts or dorsal root-evoked long-lasting reflexes, a cross-correlation analysis was performed to measure the coupling between the left and right L5 ventral bursts. The coupling was estimated by measuring the correlation coefficient at zero phase lag (center of the cross-correlogram). Positive and negative values of the correlation coefficient were representative of synchronous and out-of-phase signals, respectively. The correlation coefficient above 0.5 was taken to be indicative of synchronous activities while that below - 0.5 was taken to be indicative of a fictive locomotor episode. *Intracellular recordings:* From whole spinal cord preparations, amplitudes of IPSPs evoked in motoneurons by electrical stimulation of the ventral funiculus were measured and plotted against holding potentials. At least 22 values were collected for each motoneuron. The $E_{IPSP}$ was given by the intercept of the regression line with the *x*-axis. From slice preparations, the junction potential was corrected off-line based on the composition of the internal and external solutions used for recordings. We defined the voltage-dependent activation threshold of the $I_{NaP}$ as the membrane potential at which the slope of leak-subtracted current becomes negative. We measured the magnitude of $I_{NaP}$ as the peak of the leak-subtracted inward current during the ascending phase of the voltage command. *Immunohistochemistry:* Measurements of Nav1.6 staining were performed on initial segments from motoneurons identified as the biggest cells located in the ventral horn. Initial segments from motoneurons were identified as large linear structures labeled by Nav1.6-specific antibodies, located in the first 10 μm of the slice within the ventral horn area, and for which the beginning and the end of the structure could be clearly determined (>10 μm in length), excluding nodes of Ranvier. The mean pixel intensities of Nav1.6-specific fluorescence were measured by tracing the labeled initial segments using the multipoint line feature of the FluoView software. The measurements were repeated with similar numbers of motoneurons per animal. Because KCC2 is weakly expressed at the membrane of motoneurons during the first postnatal week (*Stil et al., 2009*) measurements of KCC2 staining were performed in the pool of motoneurons (L4-L5) by delineating a region of interest in the lateral part of the ventral horn. The same areas were considered in successive sections. For Nav1.6 and KCC2 measurements, each value was normalized to the mean value measured from sections of SCI rats on the same slide. *Western blot:* The blots were blotted with an enhanced chemiluminescence detection (Merck-Millipore). Signal intensities were measured with the image analysis software Quantity-One (BioRad).

## Drug list and solutions

Normal aCSF was used in most cases for in vitro electrophysiological recordings. To characterize $E_{IPSP}$, the aCSF solution composed of (in mM): 130 NaCl, 4 KCl, 3.75 CaCl$_2$, 1.3 MgSO$_4$, 0.58 NaH$_2$PO$_4$, 25 NaHCO$_3$, 10 D-glucose, pH = 7.4, (24–27°C) was used. To characterize $I_{NaP}$, the aCSF composed of the following (in mM): 100 NaCl, 3 KCl, 1.25 NaH$_2$PO$_4$, 1.3 MgSO$_4$, 3.6 MgCl$_2$, 1.2 CaCl$_2$, 25 NaHCO$_3$, 40 D-glucose, 10 TEA-Cl, 0.1 CdCl$_2$, pH 7.4 (32–34°C) was used. Drugs were purchased from the following sources: 1,4-Dihydro-2,6-dimethyl-5-nitro-4-(2-[trifluoromethyl]phenyl)pyridine-3-carboxylic acid methyl ester (Bay K8644, 10 μM) ; 6-cyano-7-nitroquinoxaline-2,3-dione

(CNQX, 3 µM), [(dihydroindenyl)oxy]alkanoic acid] (DIOA; 30 µM), nifedipine (20 µM), tetraethylammonium chloride (TEA; 10 mM), veratridine (60 nM) from Sigma-Aldrich; MDL28170 (30 µM) from Calbiochem; 2-amino-5-phosphonovaleric acid (AP5, 30 µM), prochlorperazine dimaleate (PCPz, 10 µM) and riluzole (5 µM) from Tocris; 4,9-Anhydrotetrodotoxin (4,9-ah-TTX, 200 nM) from Focus Biomolecules. Nifedipine, DIOA, and Bay K8644 were dissolved in dimethylsulphoxide (DMSO) and added to the ACSF (final concentration of DMSO: 0.05%). Veratridine was dissolved in ethanol and added to the ACSF (final concentration of ethanol: 0.0003%). Control experiments showed no effects of the vehicle (data not shown).

## Treatment design

On post-operative day 4–5, rats were randomly treated with a single i.p. injection of MDL28170 (60 mg/kg or 120 mg/kg) or its vehicle. These two doses of MDL28170 were chosen because the acute systemic administration of 60 mg/kg appeared to be the minimal effective dose to produce a significant inhibition of cysteine proteinase activity in the CNS of neonatal rats (*Kawamura et al., 2005*), while 120 mg/kg appeared to be more effective to inhibit calpain compared to the dose of 60 mg/kg (*Thompson et al., 2010*). The injection was performed after stable measurements of spastic motor behaviors. Regarding the pharmacokinetic profile of MDL-28170, the number of twitches was quantified as every hour up to 9 hr post-injection. The effect of the drug or its vehicle on motor responses to tail pinching was tested 1 hr post-injection.

## Statistics

No statistical method was used to predetermine sample size. Group measurements were expressed as means ± standard deviation in figures. We used Mann-Whitney test, Wilcoxon matched pairs test to compare two groups, and a one-way or a two-way ANOVA to compare more than two groups. For all statistical analyses, the data met the assumptions of the test and the variance between the statistically compared groups was similar. The level of significance was set at $p < 0.05$. Statistical analyses were performed using Prism 5.0 software (Graphpad).

# Acknowledgements

We are grateful Ronald to M Harris-Warrick and the lab members for their critical reading of the manuscript. We also thank Jérémy Verneuil for its assistance in formatting videos, Anne Duhoux for animal care and Joel Baurberg for making a miniature pressure sensor. This research was financed by a grant from Agence National de la Recherche Scientifique (CalpaSCI, ANR-16-CE16-0004), the French Institut pour la Recherche sur la Moelle épinière et l'Encéphale (IRME) and the Fondation pour la Recherche Médicale (FRM).

# Additional information

### Funding

| Funder | Grant reference number | Author |
| --- | --- | --- |
| Agence Nationale de la Recherche | ANR CalpaSCI-16-CE16-0004 | Frédéric Brocard |
| Institut pour la Recherche sur la Moelle épinière et l'Encéphale | SPV/MB/173439 | Frédéric Brocard |
| Fondation pour la Recherche Médicale | FDT20170437125 | Frédéric Brocard |

The funders had no role in study design, data collection and interpretation, or the decision to submit the work for publication.

### Author contributions

Vanessa Plantier, Data curation, Formal analysis, Validation, Investigation, Performed and analyzed most of in vitro and in vivo experiments; Irene Sanchez-Brualla, Data curation, Formal analysis, Validation, Investigation, Methodology, Performed and analyzed most of in vitro and in vivo

experiments; Nejada Dingu, Data curation, Formal analysis, Investigation, Methodology, Performed and analyzed some in vitro experiments; Cécile Brocard, Data curation, Formal analysis, Investigation, Methodology, Performed and analyzed most of biochemistry and immunohistochemistry experiments; Sylvie Liabeuf, Formal analysis, Methodology, Performed some biochemistry experiments; Florian Gackière, Data curation, Formal analysis, Methodology, Performed and analyzed most of intracellular recordings; Frédéric Brocard, Conceptualization, Formal analysis, Supervision, Funding acquisition, Validation, Investigation, Visualization, Methodology, Project administration, Designed, supervised the whole project, performed and analyzed some in vitro experiments and wrote the manuscript

### Author ORCIDs
Frédéric Brocard (ID) https://orcid.org/0000-0001-9444-9586

### Ethics
Animal experimentation: We made all efforts to minimize animal suffering and the number of animals used. All animal care and use conformed to the French regulations (Décret 2010-118) and were approved by the local ethics committee (Comité d'Ethique en Neurosciences INT-Marseille, CE Nb A1301404, authorization Nb 2018110819197361).

### Decision letter and Author response
Decision letter https://doi.org/10.7554/eLife.51404.sa1
Author response https://doi.org/10.7554/eLife.51404.sa2

## Additional files

### Supplementary files
• Transparent reporting form

### Data availability
All data generated or analysed during this study are included in the manuscript and supporting files. Source data files have been provided for all figures and figure supplements.

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
