## [Decision Letter]

**Acceptance summary:**

Your study convincingly demonstrates that calpain-I is upstream of Nav1.6-induced persistent activity and KCC2-related change in chloride potential; changes in both of these promote persistent motor neuron activity and spasms in neonatal rats. This is an interesting finding. Furthermore, inhibiting calpains restored expression of both, leading you to suggest calpains as a possible therapeutic target for spasticity. We congratulate you on a very elegant and important study.

**Decision letter after peer review:**

Thank you for submitting your article "Calpain drives the excitatory-inhibitory imbalance of motoneurons and leads to spasticity after spinal cord injury" for consideration by *eLife*. Your article has been reviewed by three peer reviewers, including Jan-Marino Ramirez as the Reviewing Editor and Reviewer #1, and the evaluation has been overseen by Ronald Calabrese as the Senior Editor. The following individual involved in review of your submission has agreed to reveal their identity: Rob Brownstone (Reviewer #3).

The reviewers have discussed the reviews with one another and the Reviewing Editor has drafted this decision to help you prepare a revised submission.

Summary:

The study by provides important insights into the mechanisms leading to spasticity after SCI. Insights gained may have general implications for TBI as well. The key finding is that calpain is a critical mediator that leads to an excitatory/inhibitory imbalance in motoneurons. Although, this finding is not entirely surprising and novel, the present study brings everything together to demonstrate this link, which involves the persistent sodium current as one of the mechanisms for increased motoneuron excitability.

Title: The authors may want to reconsider the title and make it more interesting for a general readership.

Essential revisions:

1) The authors need to provide more details with regards to the electrophysiological data. How exactly did the authors isolate the persistent sodium current. Did they use a ramp, and if so what was the time/duration and slope? The key Figure 4 doesn't contain much information, also with regards to the evoked inhibitory postsynaptic currents. It would be good to show the electrophysiological data in a separate figure, with more details. How were the neurons identified? Did the authors characterize these neurons anatomically and how do they know whether they were indeed recording from motoneurons?

2) There are a few areas where clarity could be improved. It is not clear what the number of events is, for example, in Figure 3G and 3H (and similar). How do these numbers fit with the numbers of rats as reported in the legend?

3) The notion of the excitatory-inhibitory imbalance implies a change in synaptic inputs to motoneurons. However, the main finding is mostly an upregulation of persistent Na current and consequently an increase in the excitatory of motoneurons. There is no analysis in this manuscript of how excitatory and inhibitory inputs to motoneurons are affected by spinal cord injury. Without such data, it is difficult to see how the excitatory-inhibitory balance is affected by spinal cord injury.

4) Some data presented in this study mostly confirm earlier work in adult rats. It would be very helpful to outline in the Introduction and Discussion what has been shown in adult rats and how the present study in newborn rats differs from that in adult rats.

5) There is a broad change in the excitability that it is not only limited to motoneurons. Do these changes also contribute to spasticity? It seems that the injury induces broad changes in the spinal cord and it is less clear what it is specific to motoneurons and what is the results of premotor interneurons. Clearly, the rhythmic activity seen in injured spinal cords does not reflect spasticity but rather an increased excitability of the premotor network. These aspects are mentioned in the Discussion, but it would be stronger if there were some direct analysis of the underlying mechanisms included.

6) The authors should provide some explanations regarding the difference in the time course of the emergence of spasticity in newborn versus adult rats. This study reports signs of spasticity emerging within the first week after injury while in the adult it took weeks. This raises the question if the underlying mechanisms display an age-dependent difference – some clarifications are necessary.

7) Quantitative immunohistochemistry. It is not clear what is being measured here. There can be a great deal of variability in fluorescence from slide to slide and at different depths within a section, for example. So if it is fluorescence au's that are measured, how were these controlled? Furthermore if Nav-1.6 is at the IS, then is the length/area of labeling different?

8) Bar graphs. It seems that s.e.m.'s are depicted. It is not clear why this useless number is shown. In biology, we are concerned with variability, so the variance (standard deviation) should be shown. Furthermore, while I congratulate the authors on showing much of their data, why not show the data behind these bars with scatter plots? Then we can see the variation.

---

## [Author Response]

Title: The authors may want to reconsider the title and make it more interesting for a general readership.

To make the title more relevant to a wider audience, we suggest the following title:

Calpain fosters the hyperexcitability of motoneurons after spinal cord injury and leads to spasticity.

Essential revisions:1) The authors need to provide more details with regards to the electrophysiological data. How exactly did the authors isolate the persistent sodium current. Did they use a ramp, and if so what was the time/duration and slope? The key Figure 4 doesn't contain much information, also with regards to the evoked inhibitory postsynaptic currents. It would be good to show the electrophysiological data in a separate figure, with more details. How were the neurons identified? Did the authors characterize these neurons anatomically and how do they know whether they were indeed recording from motoneurons?

Most of details are already present in the in vitro recordings and stimulation section of Material and Methods as follows:

“The main characterization of I_NaP_ was accomplished by slow ramp increase from -70 mV to -10 mV over 5 s, slow enough (12 mV/s) to prevent transient sodium channel opening.”

“Motoneurons were identified by the antidromic response to stimulation of the ventral roots. Stimulation of the ventral funiculus usually induced inhibitory postsynaptic potentials (IPSPs) in the presence of 2-amino-5-phosphonovaleric acid (AP5, 30–100 μM) and 6-cyano-7-nitroquin-oxaline-2,3-dione (CNQX, 3–10 μM). IPSPs were recorded at various holding potentials (500 ms-long current pulses).”

“Motoneurons were visually identified with video microscopy (Nikon Eclipse E600FN) coupled to infrared differential interference contrast, as the largest cells located in layer IX”

To help the reader, we added some details of the protocol in the corresponding figure legend. We also illustrated the voltage ramp protocol used for evoking I_NaP_ and the stimulus artifact for evoking IPSPs. We have not made a separate figure with the electrophysiological data because we believe their presence in Figure 4 is more useful.

2) There are a few areas where clarity could be improved. It is not clear what the number of events is, for example, in Figure 3G and 3H (and similar). How do these numbers fit with the numbers of rats as reported in the legend?

To clarify this point we added the following text in each legend when it was appropriate:

“Group means quantification of: events per rat detected over time windows of 10–40 ms and 500–15,000 ms post-stimulus for SLR and LLR, respectively.”

More details were also added in Material and Methods as follows:

“PSTHs were constructed from 5 consecutive rectified responses. We computed the transient short latency (SLR) and long-lasting reflexes (LLR) over time windows of 10–40 ms and 500–15,000 ms post-stimulus, respectively. Time windows were discretized into 20 ms-bins. For each bin, we calculated the number of events using a threshold search on ClampFit, the result was added in the corresponding time window. For each animal, we obtained one value for the SLR and one for the LLR, that were used for statistical analysis.”

3) The notion of the excitatory-inhibitory imbalance implies a change in synaptic inputs to motoneurons. However, the main finding is mostly an upregulation of persistent Na current and consequently an increase in the excitatory of motoneurons. There is no analysis in this manuscript of how excitatory and inhibitory inputs to motoneurons are affected by spinal cord injury. Without such data, it is difficult to see how the excitatory-inhibitory balance is affected by spinal cord injury.

We recognize that, as the reviewers said, to talk about the excitatory-inhibitory balance we should have studied the synaptic input onto motoneurons, which was part of our experiments concerning inhibition (See Figure 4K). However, we agree that we did not investigate changes in the excitatory inputs to motoneurons. What we are describing, instead, is the hyperexcitability of motoneurons related to the increase of the persistent sodium current, which is at the origin of spasticity and spasms that we observe. Therefore, we replaced throughout the manuscript all mentions to the “excitatory-inhibitory imbalance” by “hyperexcitability”.

4) Some data presented in this study mostly confirm earlier work in adult rats. It would be very helpful to outline in the Introduction and Discussion what has been shown in adult rats and how the present study in newborn rats differs from that in adult rats.

To meet the request we distinguished in the Introduction studies made in adults from those in newborns. To emphasize the novelty of our study compared to previous ones we rewrote the first paragraph of the Discussion as follows:

“Numerous studies on adult animal models delineate the hyperexcitable state of motoneurons as one of the causes of spasticity, presumably because of an increase of INaP and a decrease of KCC2 expression. The present study performed in neonatal rats identifies the activation of calpains as the upstream mechanism of the hyperexcitability of motoneurons after SCI. Calpain-mediated cleavage of Nav channels and KCC2 up- and down-regulates INaP and inhibition, respectively, in lumbar motoneurons. The reduction of calpain activity normalizes INaP and the strength of inhibitory transmission, and thereby alleviates spasticity. Altogether, these findings uncover a new cellular mechanism contributing to spasticity and provide novel therapeutic targets…”

5) There is a broad change in the excitability that it is not only limited to motoneurons. Do these changes also contribute to spasticity? It seems that the injury induces broad changes in the spinal cord and it is less clear what it is specific to motoneurons and what is the results of premotor interneurons. Clearly, the rhythmic activity seen in injured spinal cords does not reflect spasticity but rather an increased excitability of the premotor network. These aspects are mentioned in the Discussion, but it would be stronger if there were some direct analysis of the underlying mechanisms included.

We agree with the reviewers, and as we already mentioned in the discussion, the hyperexcitability is likely not limited to motoneurons. As the reviewer pointed out, in some of our experiments we found a rhythmic activity in the injured spinal cord in vitro, which is most likely caused by premotor neurons and which is implicated in generating muscle spasms (Husch et al., 2012; Bellardita et al., 2017; Lin et al., 2019). We are currently working on the mechanisms underlying this phenomenon in our model. However, it is too premature to add these preliminary results in the following study, since they will be the object of an independent paper. Therefore, we think that at this point we can only discuss our data.

6) The authors should provide some explanations regarding the difference in the time course of the emergence of spasticity in newborn versus adult rats. This study reports signs of spasticity emerging within the first week after injury while in the adult it took weeks. This raises the question if the underlying mechanisms display an age-dependent difference – some clarifications are necessary.

We are grateful to the reviewers for raising this interesting point. We have described the onset of spasticity in newborn rats, which is surprisingly much faster than the onset in adult rats. We do not know exactly why is that so, but we discussed two non-exclusive mechanisms as follows:

“The emergence of spasticity is surprisingly faster than that of adult animals, which appears several weeks after the injury. A more active calpain-mediated proteolysis during early development may explain this age-dependent discrepancy. Indeed, calpain proteolysis is necessary for the pruning of axons during development (Yang et al., 2013) and the natural calpain inhibitory peptide, calpastatin, increases slowly in the CNS during the first weeks of life (Li et al., 2009). The lower presence of this inhibitor in newborn rats may explain why hyperexcitability after a neonatal SCI developed faster. Also at birth, rat motoneurons overexpress NMDA receptors (Vinay et al., 2000) which have been demonstrated to be efficient in activating calpains in pathological conditions (Zhou et al., 2012). In sum, the low expression of calpastatin combined with a high expression of NMDA receptors may contribute to a fast increase of motoneuronal excitability after a SCI in newborn rats.”

The following references were added in the manuscript:

Li et al., (2009). Calpain 1 and Calpastatin expression is developmentally regulated in rat brain. Exp Neurol. 2009 Dec;220(2):316-9.doi:10.1016/j.expneurol.2009.09.004. Epub 2009 Sep 12.

Vinay et al. (2000) Perinatal development of lumbar motoneurons and their inputs in the rat. Brain Res Bull 53(5):635-47.

Yang et al. (2013) Regulation of axon degeneration after injury and in development by the endogenous calpain inhibitor calpastatin. Neuron 80(5):1175-89.

7) Quantitative immunohistochemistry. It is not clear what is being measured here. There can be a great deal of variability in fluorescence from slide to slide and at different depths within a section, for example. So if it is fluorescence au's that are measured, how were these controlled? Furthermore if Nav-1.6 is at the IS, then is the length/area of labeling different?

Sections from treated and untreated SCI rats were mounted on the same slides and processed simultaneously. We used identical settings and quantification of staining intensities were performed using the FluoView Sofware (version 5, Olympus). Measurements were performed on AISs of biggest cells located in the first 10 µm of the slice within the ventral horn region, and for which the beginning and the end of the structure could be clearly determined (>10 μm in length), excluding nodes of Ranvier. The mean pixel intensities of Nav1.6-specific fluorescence were measured by tracing the labelled AISs using the multipoint line feature of the FluoView software, on the optical section where the intensity of fluorescence was optimal. Each value representing the intensity of fluorescence averaged to the length of the AISs was then normalized to the mean value measured from sections of untreated SCI rats on the same slide. Analysis were performed on a similar number of motoneurons per animal. This information is originally described in data analysis section of Materials and methods.

8) Bar graphs. It seems that s.e.m.'s are depicted. It is not clear why this useless number is shown. In biology, we are concerned with variability, so the variance (standard deviation) should be shown. Furthermore, while I congratulate the authors on showing much of their data, why not show the data behind these bars with scatter plots? Then we can see the variation.

Due to the large number of samples and the small size of histograms, illustrating data with scatter plots would make the histograms unreadable. However to meet the concern we replaced all SEM by SD in figures. All SDs are also in data sources. The statistics section in the Materials and Methods was amended accordingly.